# Mathematical properties of optimal fluxes in cellular reaction networks at balanced growth

**Hugo Dourado**[1]*, **Wolfram Liebermeister**[2], **Oliver Ebenhöh**[3], **Martin J. Lercher**[1]

**1** Institute for Computer Science and Department of Biology, Heinrich-Heine Universität, Düsseldorf, Germany, **2** Université Paris-Saclay, INRAE, MaIAGE, Jouy-en-Josas, France, **3** Quantitative and Theoretical Biology, Heinrich-Heine Universität, Düsseldorf, Germany

\* hugo.dourado@hhu.de

**Data Availability Statement:** All data and codes are available in the S1 File.

**Funding:** This work was funded by a Volkswagenstiftung "Life?" grant to MJL, and by

## Abstract

The physiology of biological cells evolved under physical and chemical constraints, such as mass conservation across the network of biochemical reactions, nonlinear reaction kinetics, and limits on cell density. For unicellular organisms, the fitness that governs this evolution is mainly determined by the balanced cellular growth rate. We previously introduced growth balance analysis (GBA) as a general framework to model and analyze such nonlinear systems, revealing important analytical properties of optimal balanced growth states. It has been shown that at optimality, only a minimal subset of reactions can have nonzero flux. However, no general principles have been established to determine if a specific reaction is active at optimality. Here, we extend the GBA framework to study the optimality of each biochemical reaction, and we identify the mathematical conditions determining whether a reaction is active or not at optimal growth in a given environment. We reformulate the mathematical problem in terms of a minimal number of dimensionless variables and use the Karush-Kuhn-Tucker (KKT) conditions to identify fundamental principles of optimal resource allocation in GBA models of any size and complexity. Our approach helps to identify from first principles the economic values of biochemical reactions, expressed as marginal changes in cellular growth rate; these economic values can be related to the costs and benefits of proteome allocation into the reactions' catalysts. Our formulation also generalizes the concepts of Metabolic Control Analysis to models of growing cells. We show how the extended GBA framework unifies and extends previous approaches of cellular modeling and analysis, putting forward a program to analyze cellular growth through the stationarity conditions of a Lagrangian function. GBA thereby provides a general theoretical toolbox for the study of fundamental mathematical properties of balanced cellular growth.

## Author summary

Mathematical models are an important tool to understand and predict the complex behavior of biological cells. This behavior is driven by nonlinear physical constraints that cannot be captured entirely in the prevalent modeling frameworks, which rely on simplified linear optimizations. The next generation of more realistic cell models will depend on an

the German Research Foundation through grant CRC 1310 to MJL, and, under Germany's Excellence Strategy, grant EXC 2048/1 (Project ID: 390686111) to OE and MJL. The funders had no role in study design, data collection and analysis, decision to publish, or preparation of the manuscript.

**Competing interests:** The authors have declared that no competing interests exist.

efficient mathematical formulation for the corresponding nonlinear optimization problem that facilitates the analytical study and numerical simulation of large models. Here, we present a succinct formulation for this nonlinear problem, and we derive the analytical properties of fluxes at optimal growth. We also show how these analytical properties can be understood in terms of economics and control theory, where they expose trade-offs related to the allocation of proteins.

## Introduction

A core feature of microbial cells is self-replication—their ability to build a complete, identical cell exclusively out of the chemical compounds found in the environment. If a population of asynchronously replicating microbial cells grows exponentially in a constant environment, its self-replication can often be assumed to result from balanced growth, a non-equilibrium steady state in which every cellular component accumulates at the same rate in proportion to its total amount [1]. For non-interacting microbes in a constant environment, the balanced growth rate is equivalent to fitness [2].

The cellular composition is thus often interpreted as an approximate solution to a problem of optimal allocation, driven by natural selection. Accordingly, theoretical methods estimating the optimal allocation are used as a reference to understand cellular composition *in vivo* [3–8].

At the whole-cell level, a mechanistic understanding of the quantitative principles that shape cellular balanced growth has been approached predominantly through methods collectively classified as constraint-based modeling (CBM). CBM approaches define a solution space of feasible cellular states (usually defined by reaction fluxes) based on simple, mechanistic constraints. The predominant constraint in CBMs is flux balance, encoded through a linear system of equations that constrain the space of allowed reactions fluxes $\mathbf{v}$ [9, 10],

$$\mathbf{S}\,\mathbf{v} = \mathbf{0}. \tag{1}$$

Here, $\mathbf{v}$ is a vector of reaction fluxes, i.e., reaction rates in units of $[\text{moles}][\text{time}]^{-1}[\text{volume}]^{-1}$. Each row of the stoichiometric matrix $\mathbf{S}$ corresponds to one metabolite, while each column corresponds to a metabolic reaction, with entries listing the corresponding stoichiometric coefficients of substrates (negative integers) and products (positive integers).

Thermodynamics and physiological limits—such as a limited nutrient uptake capacity—are typically approximated through fixed upper and/or lower bounds on the modeled fluxes $\mathbf{v}$ [11]. The most widely used CBM approach, Flux Balance Analysis (FBA) [11, 12], obtains plausible physiological states by optimizing some objective function over the feasible flux vectors. Frequently, the objective function is the flux through a hypothetical biomass reaction $v_{\text{bio}}$, which mimics the accumulation of precursors for macromolecules and the consumption of energy for their assembly during growth.

Resource Balance Analysis (RBA) and metabolism and expression models (ME-models)—which are also based on optimization under constraints—go beyond FBA by aiming to model metabolism in its most general sense, with the ultimate goal of representing all chemical reactions that occur in a living organism [8, 13]. In contrast to FBA, these methods take into account the burden of producing the macromolecules (proteins and RNA) required for catalyzing each flux. They approximate the corresponding kinetic rate laws as linear relations between fluxes and the concentration of their catalysts, ignoring the dependence on reactant concentrations (except for dependencies on extracellular concentrations, which serve as model parameters).

All widely used CBMs [8, 11, 13, 14] are formulated as linear optimization problems, which can be solved efficiently even for genome-scale models with thousands of reactions. Accordingly, they are currently the most efficient tools to predict and understand realistic cellular models. However, by construction, these linear methods cannot capture the potentially complex nonlinear relationship between biochemical reaction fluxes—and hence cellular growth — and the concentrations of reactants involved as substrates and products. Instead of accounting for nonlinear kinetics, these methods rely on linear, phenomenological assumptions.

Nonlinear CBMs [6, 15–19] account for constraints such as nonlinear kinetic rate laws, linking the concentration of metabolites to reaction fluxes. This link means that the metabolite concentrations are now an output of the model instead of an input. Molenaar et al. [6] introduced "self-replicator" models that maximize the cellular growth rate, with reaction fluxes that are limited by fundamental physiological constraints including mass conservation, nonlinear rate laws, and limited protein density. Importantly, these models are completely self-contained, in the sense that in order to grow and self-replicate, all of a model's individual components have to be produced explicitly by the model itself. Instead of using a phenomenological "biomass reaction", the constrained optimization of growth predicts the detailed cell composition, and all possible trade-offs in resource allocation can be accounted for from first principles.

Similar to RBA and ME models, self-replicator models include a "ribosome" reaction that produces the necessary proteins. The proteins can be classified into three categories: transport proteins in the cell surface, which exchange mass with the environment; enzymes, which catalyze internal metabolic reactions; and the ribosome itself, which catalyzes the internal protein production, and which for simplicity is assumed to be composed only of proteins. The study of models of this type relies on the numerical solution of nonlinear optimizations: while it in principle accommodates models with any number of reactions, actual presented models have small, highly simplified reaction networks [6, 15–19].

We have previously formalized a general framework for modeling and analyzing nonlinear CBMs, an approach we termed growth balance analysis (GBA) [4] (Fig 1). GBA models are based on the self-replicator scheme, but instead of considering a fixed protein concentration, they consider a fixed combined mass density of all intracellular components, including also metabolites. Optimal cellular resource allocation, as predicted by GBA models, emerges exclusively from quantitative biochemical and physical principles, including the intrinsic nonlinear nature of the underlying reaction kinetics. In general, the optimization of nonlinear models is a non-convex problem, frequently hampered by the existence of multiple local optima [20]. Several studies have explored ad-hoc analytical solutions to convex, minimal nonlinear cell models consisting of up to three cellular reactions. Despite their simplicity, simulations with these schematic models are qualitatively consistent with the experimentally observed behaviour of actual cells [6, 15–19].

The optimization of large nonlinear CBMs, such as GBA models, is still an open problem for numerical methods [20]. Thus, in previous work, we proposed a different approach to the analysis of GBA models—instead of looking for the optimal state of a GBA model with numerical methods, we ask: what are the analytical properties of this optimal state? We named the equations specifying these analytical properties the *balance equations* of the corresponding GBA model [4]. If we further assume, as most CBM approaches do [6, 8, 11, 13–19], that cells are at an optimal state (or at least close to one), then the balance equations become useful tools to estimate and understand the resource allocation in actual biological cells. We derived the balance equation for each reactant in a GBA mode and applied these equations to successfully predict the protein allocation into the ribosome of both *E. coli* and yeast across various growth

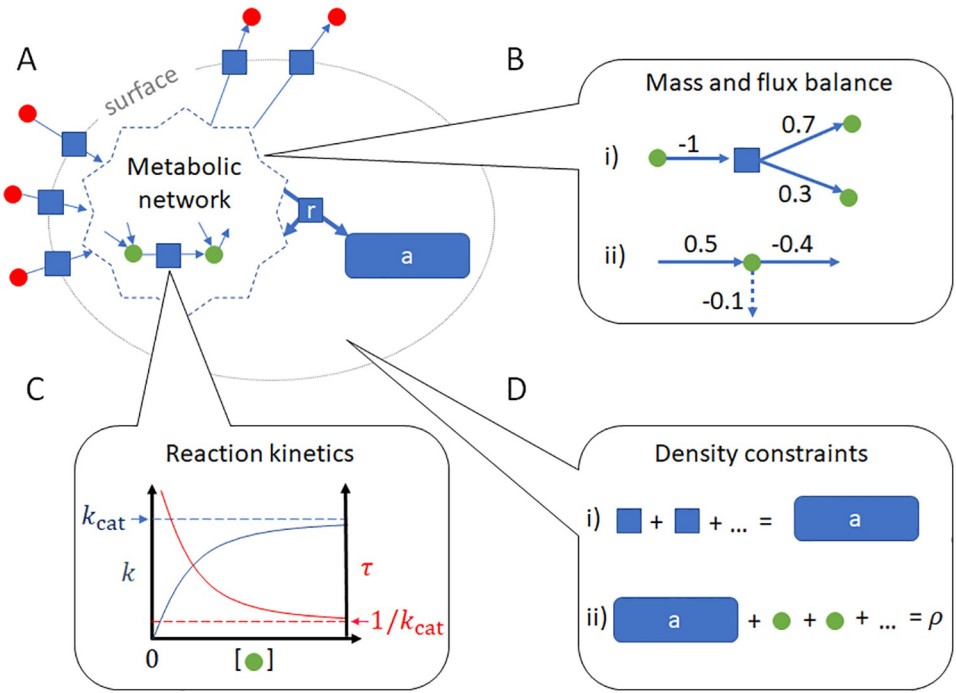

**Fig 1. Constraints in a GBA model. A)** In a GBA model, a cell exchanges external reactants (red circles) via transporters (blue squares at the cell surface); converts internal reactants (green circles) via enzymatic reactions (blue squares inside the metabolic network); and produces all proteins catalyzing the reactions (blue rectangle "a") via a ribosome reaction "r". The ribosome reaction consumes and returns metabolites to the metabolic network. In its strict sense, the metabolic network comprises the conversion of small molecules into energy and precursors for macromolecules. A model may also describe metabolism in its more general sense, including other enzymatic reactions such as those for DNA replication and transcription. **B)** All reactions in the model must conserve mass, a concept that comprises (i) mass balance within reactions: one unit of mass consumed (-1) results in one unit of mass produced (+1); and (ii) flux balance of reactant production and consumption, including the dilution by growth of all components (dashed arrow). **C)** Each reaction flux is catalyzed by a specific protein with turnover time $\tau$ (or equivalently, turnover rate $k = 1/\tau$). $\tau$ is determined by kinetic rate laws and depends on the concentrations of reactants involved in the reaction; $k = 1/\tau$ has a maximal value $k_{\text{cat}}$. **D)** Two basic density constraints govern the cellular interior: (i) the density of proteins "a", and (ii) the total density $\rho$, which is the sum of all protein and metabolite concentrations.

conditions [4]. The application of these balance equations to estimate complete cellular states is however limited; in our original derivation [4], the stoichiometric matrix **S** of interest is assumed to have full column rank, which is the case for all optimal states of GBA models when one restricts **S** to the columns corresponding to *active* reactions (i.e., those with non-zero flux) [4, 21, 22]. The active reactions at optimal growth form an Elementary Growth Mode (EGM) [23], and they represent an Elementary Flux Mode (EFM) [21, 22] of a related FBA problem [4]. Unfortunately, however, the optimal choice of this EGM/EFM and its constituting active reactions is not known *a priori* for large-scale models, and cannot be explained by our previous analytical study [4].

Below, we generalize our previous analytical study of GBA models by deriving the analytical properties of each reaction at optimal balanced growth, now also accounting for models with column-rank-deficient stoichiometric matrices—i.e., biochemical networks with alternative pathways. These analytical properties can be seen as generalized balance equations, explaining from first principles the optimal resource allocation strategy for each reaction in a cell. In particular, they explain from first principles the exact mathematical condition determining whether a reaction is active or not in an optimal growth state. We then interpret these balance

equations in terms of marginal costs and benefits of reactions with respect to their influence on growth, and quantify how changes in the model parameters and external conditions control the optimal growth rate.

## Results

We first present the notation and mathematical definitions for growth optimization, including an objective function and constraints. We then reformulate the problem in terms of flux fractions as the only free variables, which greatly simplifies the subsequent analytical study. Finally, we explore the consequences that emerge from the necessary optimality conditions in terms of economics and control theory, and discuss their biological significance.

### Growth modeling

We define a GBA model as the triple of model parameters $(\mathbf{M}, \boldsymbol{\tau}, \rho)$. The matrix $\mathbf{M}$ describes the mass fractions of internal reactants consumed and produced by each reaction; $\boldsymbol{\tau}$ is a vector of catalytic turnover times for all reactions, where each is a function of internal reactant concentrations $\mathbf{c}$ and possibly also external concentrations $\mathbf{x}$ (assumed to be fixed and given); and $\rho$ is the combined mass concentration of all internal components. In the following paragraphs, we provide more detailed descriptions of the model constituents $\mathbf{M}$, $\boldsymbol{\tau}$, and $\rho$. Here and below, we use the term "reaction" to also encompass transport processes across the cell surface, which are "catalyzed" by transporter proteins or protein complexes.

The matrix $\mathbf{M}$ was first introduced in [4]. It is constructed from the stoichiometric matrix $\mathbf{S}$ for the total, closed system, i.e., including rows for external reactants. We add a column "r" for the ribosome reaction that produces all cellular protein, as well as a row "a" corresponding to the total concentration of all proteins in mass units. We now first convert all entries to masses, by multiplying each row with the corresponding molecular mass. Because of mass balance, each column must then sum to 0. We next normalize each column such that the sum of its negative entries equals −1 and the sum of its positive entries equals +1. Now the entries correspond to the mass fractions of each reactant (rows) going into and out of each reaction (columns), as illustrated for the example in Fig 1B. Finally, we reduce the normalized matrix to a matrix for an open system, by dropping all rows for reactants external to the modeled cell. For the remaining internal reactants, we will assume a quasi-steady state and thus enforce mass conservation.

As illustrated in Fig 2, to simplify the notation for the following theoretical development, we partition the columns of $\mathbf{M}$ (indexed together by $j$) into index sets for reaction types: $s$ for transport processes across the cell surface; $e$ for internal enzymatic reactions; and r for the ribosome reaction, which is the only one that produces protein. We partition the rows of internal reactants (indexed together by $i$) into indices $m$ for metabolites and a for total protein. We use the term "metabolites" in its more general sense, referring to any molecule in the cell that is not a protein. We distinguish vectors by using boldface, and vector components by using italics with the appropriate upper or lower index, e.g., $\mathbf{c}$ is the column vector of all internal reactant concentrations, $c^i$ are its components, and we use a lower index to indicate the components $c_i$ of the row vector $\mathbf{c}^\top$.

The ribosome reaction represents the last step in protein synthesis, and is assumed to be catalyzed by a "ribosome" consisting entirely of protein. We here ignore the RNA components of the ribosome for simplicity, but it is possible to extend the modeled ribosome to a more realistic RNA-protein complex. In addition, the enzymatic reactions ($e$) could be extended so that they include details of protein translation that occur before the last, "ribosome" step ($r$) [5]. Note that nonlinear genome-scale GBA models can be created from existing linear genome-

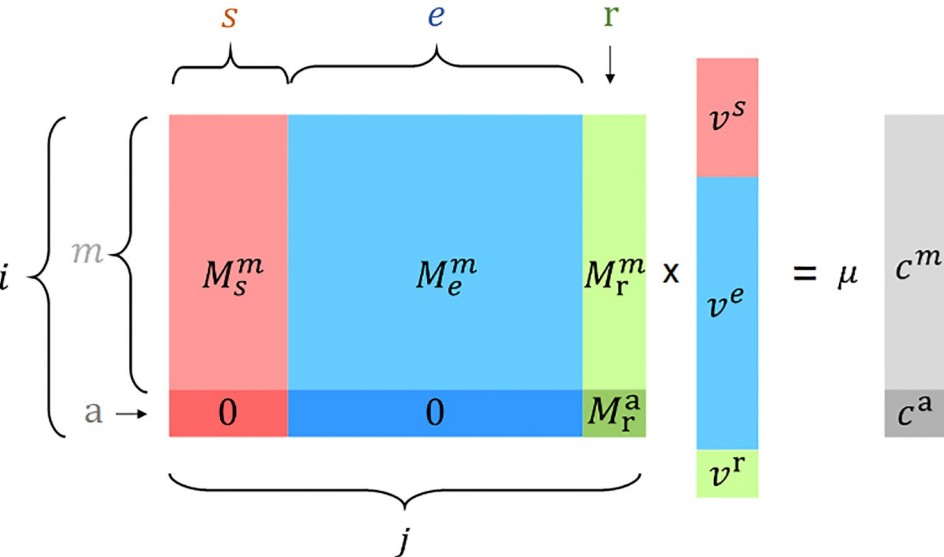

**Fig 2. Schematic overview of the mass conservation constraint.** $\mathbf{M}\,\mathbf{v} = \mu\mathbf{c}$, determined by the mass fraction matrix $\mathbf{M}$, the column vector of mass fluxes $\mathbf{v}$, the growth rate $\mu$, and the column vector of internal reactants mass concentrations $\mathbf{c}$. The indices indicate partitions according to the type of reaction (columns of $\mathbf{M}$, $\mathbf{v}$) or reactant (rows of $\mathbf{M}$, $\mathbf{c}$). The index $i = (m, \mathrm{a})$ correspond to rows for internal reactants, comprising rows $m$ for metabolites, and a row "a" for the total mass concentration of all proteins. The index $j = (s, e, \mathrm{r})$ correspond, respectively, to transport proteins, enzymes, and the ribosome. We also use the index $l$ for all reactions when necessary. Note the row "a" of $\mathbf{M}$ has only one nonzero entry $M_{\mathrm{r}}^{\mathrm{a}}$, corresponding to the mass fraction of protein produced by the ribosome reaction r. Different colors indicate three different types of reactions: red (transporters), blue (enzymes), green (ribosome); and two types of reactants: light gray (metabolites), dark gray (total protein), resulting in six partitions of $\mathbf{M}$.

scale models, by extending their stoichiometric matrix $\mathbf{S}$ with the addition of a ribosome reaction, normalizing it to $\mathbf{M}$ with the molecular masses, and adding the kinetic rate laws $\boldsymbol{\tau}$ and density $\rho$ (see below).

We assume that every reaction represented by a column $j$ in $\mathbf{M}$ is catalyzed by a protein or protein complex with concentration $p^j$—specifically, a transporter ($s$), an enzyme ($e$), or the ribosome (r). The corresponding flux $v^j$ is assumed to be proportional to $p^j$, expressed as $v^j = p^j/\tau^j(\mathbf{c}, \mathbf{x})$. Thus, $\tau^j$ is a function defined as the inverse of the usual metabolite-dependent factor in kinetic rate laws. $\tau^j$ must have a negative value when the flux is negative. Fig 1C shows the relationship between turnover time $\tau$ and turnover number $k_{\mathrm{cat}}$. S1 Text provides a basic discussion of rate laws and the necessary kinetic parameters. $\mathbf{c} = (c^m, c^{\mathrm{a}})^\top$ is the vector of internal reactant concentrations, comprising all metabolite concentrations $c^m$ as well as $c^{\mathrm{a}}$, the combined mass concentration of all proteins. Hence, each $\tau^j$ may depend not only on the concentrations of the substrates and products of the corresponding reaction, but also on inhibitors and regulatory metabolites not involved in the turnover itself. The transport processes $s$ are the only reactions whose rate laws may depend on the external concentrations $\mathbf{x}$.

Note that in accordance with the normalization of $\mathbf{M}$, all concentrations of proteins $p^j$ and reactants $c^i$ throughout this work are in units of $[\mathrm{mass}][\mathrm{volume}]^{-1}$. Fluxes ($[\mathrm{mass}][\mathrm{volume}]^{-1}[\mathrm{time}]^{-1}$) and the kinetic parameters must then also be expressed in mass units, e.g., Michaelis constants $K_{\mathrm{m}}$ in $[\mathrm{mass}][\mathrm{volume}]^{-1}$ and turnover numbers $k_{\mathrm{cat}}$ in product mass per protein mass per time, resulting in $[\mathrm{time}]^{-1}$.

$\rho$, the final constituent of GBA models, is the sum of all internal concentrations. We assume $\rho$ to be constant, which is consistent with experimental data on *E. coli* across growth conditions and even across the cell cycle [24–26]. The mass balances exploited for the normalization of $\mathbf{M}$

mean that all reactants involved in reactions must be accounted for in the model and hence be included in the value of $\rho$; e.g., in a realistic model water is a reactant in many reactions, so $\rho$ corresponds in this case to the total cellular density (or buoyant density). Simplified models may instead include only dry mass components, so that both $\mathbf{M}$ and $\rho$ consider only these.

Mass conservation implies that in the mass fraction matrix $\mathbf{M}$, each column sum $\gamma_j := \sum_i M_j^i$ is zero if it involves only the consumption and production of internal reactants (indices $e$, r). In contrast, transport reactions (with indices $s$), which bring mass into and out of the modeled system, do not conserve mass, resulting in the equations

$$
\begin{aligned}
\gamma_r &= 0 \\
\gamma_e &= 0 \\
\gamma_s &\neq 0 \quad .
\end{aligned}
\tag{2}
$$

The property (2) guarantees mass conservation within reactions, an information that is not always fully encoded in the stoichiometric matrix $\mathbf{S}$, as many models ignore common reactants such as water (see discussion in S1 Text). While external reactants have no corresponding rows in $\mathbf{M}$, their concentrations $\mathbf{x}$ may influence the turnover times $\tau^s$ of transporters. We present examples of GBA models in S1 Text and R code for their numerical optimization in S1 File.

We are interested in the cellular physiology, defined through the concentration vectors $\mathbf{c}$, $\mathbf{p}$ and the vector of reaction fluxes $\mathbf{v}$, at balanced growth. For a given model specified by $(\mathbf{M}, \boldsymbol{\tau}, \rho)$ and a given environment specified by $\mathbf{x}$, balanced growth at the instantaneous rate $\mu$ is specified by the following constraints:

$$
\mathbf{M}\mathbf{v} = \mu\mathbf{c} \qquad \text{(mass conservation in balanced growth)} \tag{3}
$$

$$
\mathbf{p} = \mathbf{v} \odot \boldsymbol{\tau}(\mathbf{c}, \mathbf{x}) \qquad \text{(reaction kinetics)} \tag{4}
$$

$$
c^a = \sum_j p^j \qquad \text{(definition of total protein density } c^a) \tag{5}
$$

$$
\rho = \sum_i c^i \qquad \text{(constant cellular density)} \tag{6}
$$

$$
\mathbf{c} \geq \mathbf{0} \qquad \text{(non–negative reactant concentrations)} \tag{7}
$$

$$
\mathbf{p} \geq \mathbf{0} \quad , \qquad \text{(non–negative protein concentrations)} \tag{8}
$$

where $\odot$ indicates element-wise multiplication. We say that any state $(\mathbf{c},\mathbf{p},\mathbf{v})$ satisfying Eqs 3–8 with growth rate $\mu > 0$ is a Balanced Growth State (BGS) for the model specified by $(\mathbf{M}, \boldsymbol{\tau}, \rho)$ and the environment specified by $\mathbf{x}$. The Optimal Growth State (OGS) is the BGS resulting in the maximal growth rate $\mu$. It can be shown that any OGS must always use a minimal subset of active reactions, i.e., growth becomes impossible if one of the active reactions is deactivated without simultaneously activating other reactions [4, 21, 22]. We term BGSs that use such minimal subsets of reactions Elementary Growth States (EGSs) [4]. Each EGS corresponds to an Elementary Flux Mode (EFM) [27] of the "linearized" version of the balanced growth problem with fixed concentrations $\mathbf{c}$ [4]; in that case, $\boldsymbol{\tau}(\mathbf{c}, \mathbf{x})$ also have fixed values and all Eqs 3–8 become linear. EGSs are specific instances of Elementary Growth Modes (EGMs) [23], sets of states using the same minimal set of active reactions.

The model's balanced growth property is captured by the right hand side of Eq 3. We assume that the growth rate is always positive, $\mu > 0$. Thus, for internal nodes with non-zero concentration ($c^i \neq 0$), there is a necessary mass flow to offset the dilution through the associated volume growth at rate $\mu$ [28]. Note that the total protein concentration $c^a$ defined by Eq 5 has no fixed value, but is constrained by the concentration of enzymes and transporters required to sustain the reaction fluxes, by a row "a" in Eq 3 that specifies mass conservation in balanced growth, and by the total density constraint in Eq 6. We summarize our assumptions about proteins as the following: i) all proteins have the same amino acid composition (determined by the entries in the column $\mathbf{M}_r$) and are produced by the ribosome following identical kinetics; ii) the total protein concentration $c^a$ is defined as the sum of all protein mass concentrations $\mathbf{p}$ by Eq 5; iii) $c^a$ relates to the ribosome flux $v^r$ and growth rate $\mu$ via the row "a" in the mass conservation constraint given by Eq 3 ($M_r^a v^r = \mu c^a$); and iv) $c^a$ also relates to the density constraint via the sum in Eq 6. From our modeling perspective, we might think of "protein" as being produced by the ribosome and instantly distributed across all reactions such that each individual protein catalyst (transporter, enzyme, or ribosome) maintains its concentration in balanced growth.

Below, we will be interested in the analytical properties of the OGS for a given model ($\mathbf{M}$, $\boldsymbol{\tau}$, $\rho$) and environment $\mathbf{x}$. From Eqs 3–5, we see that the variables ($\mathbf{c}$, $\mathbf{p}$, $\mathbf{v}$) are highly interdependent. The above formulation does not lend itself to expressing $\mu$ as an explicit function of these variables, which makes it not ideal for numerical or analytical studies. If one can find a mathematically equivalent formulation based on fewer, independent variables, then this would facilitate the use of the KKT conditions, analogous to how generalized coordinates facilitate the solution of problems in Lagrangian mechanics [29]. Thus, we next focus on a corresponding reformulation of the optimization problem. This formulation will apply to all BGSs, and only later we will use it to examine OGSs.

## A reformulation in terms of flux fractions f

Our guiding thought below is that there can be a correspondence between cell states at different growth rates, which can be expressed in the form of scaling relations. These scaling relations extend the mass fraction scaling of $\mathbf{M}$ to fluxes and concentrations. Specifically, we define *biomass fractions*

$$\mathbf{b} := \frac{\mathbf{c}}{\rho} \quad \text{(adimensional)} \tag{9}$$

(equivalent to $\mathbf{c} = \rho \mathbf{b}$, since $\rho > 0$), which express concentrations as fractions of the total cellular density; and we define *flux fractions*

$$\mathbf{f} := \frac{\mathbf{v}}{\mu\rho} \quad \text{(adimensional)} \tag{10}$$

(equivalent to $\mathbf{v} = \mu\rho \mathbf{f}$, since we assume $\mu\rho > 0$), which express fluxes as fractions of the net mass uptake — i.e., the net growth — of the cell, $\mu\rho$. Thus, each flux fraction $f^j$ describes the activity of reaction $j$ relative to the total cellular mass production. We note that the flux fractions $\mathbf{f}$ may in principle assume any real value, in the same way the fluxes $v^j$ do, including negative values when the corresponding reactions are running backwards (in which case $\tau^j < 0$). The reversibility of reactions is not an input, but an emergent output pattern here, due to the sign of each turnover time $\tau^j$, as we discuss below.

Importantly, Eq 3 describing mass conservation in balanced growth does not depend explicitly on $\mu$ anymore when written in terms of $\mathbf{f}$ and $\mathbf{b}$:

$$\mathbf{M}\,\mathbf{f} = \mathbf{b} \quad . \tag{11}$$

This equation also implies that the mass fractions $\mathbf{b}$ are uniquely determined by the flux fractions $\mathbf{f}$, independently of $\mu$. Conveniently, this unique dependence also means that we can express the turnover times as functions of only $\mathbf{f}$ and the fixed parameters $\rho$, $\mathbf{M}$, and $\mathbf{x}$:

$$\boldsymbol{\tau} = \boldsymbol{\tau}(\mathbf{c}, \mathbf{x}) = \boldsymbol{\tau}(\rho\,\mathbf{M}\,\mathbf{f}, \mathbf{x}) \quad . \tag{12}$$

In the following discussion, we mostly focus on the dependence of $\boldsymbol{\tau}$ on $\mathbf{f}$, and for simplicity of notation we do not state the dependence of $\boldsymbol{\tau}$ on the fixed parameters ($\rho$, $\mathbf{M}$, $\mathbf{x}$) explicitly. Importantly, $\boldsymbol{\tau}$ does not depend explicitly on $\mu$, which otherwise would cause a recursion problem when further expressing the growth rate $\mu$ in terms of only $\mathbf{f}$ and $\boldsymbol{\tau}(\mathbf{f})$, as we will see below. The resulting dimensionality reduction of the solution space not only simplifies the analytical considerations below, but also potentially accelerates numerical optimizations [30].

From Eqs 4 and 10, we obtain the expression for protein concentrations in terms of $\mathbf{f}$, $\mu$, and $\rho$,

$$\mathbf{p} = \mu\,\rho\,\mathbf{f} \odot \boldsymbol{\tau}(\mathbf{f}) \quad . \tag{13}$$

The combined mass fraction of all proteins in the cell, $b^{\mathrm{a}}$, is the sum of all $\mathbf{p}$ in the last equation, divided by $\rho$:

$$b^{\mathrm{a}} = \mu\,\mathbf{f}^{\top}\boldsymbol{\tau}(\mathbf{f}) \quad . \tag{14}$$

Thus, we can calculate the total protein mass fraction during balanced growth from $\mu$ and $\mathbf{f}$, based only on reaction kinetics. However, through Eq 11, the same total protein mass fraction is also related to $\mathbf{f}$ through the corresponding row "a" in $\mathbf{M}$:

$$b^{\mathrm{a}} = \mathbf{M}^{\mathrm{a}}\,\mathbf{f} = M_{\mathrm{r}}^{\mathrm{a}}f^{\mathrm{r}} \quad , \tag{15}$$

where $\mathbf{M}^{\mathrm{a}}$ is the row of $\mathbf{M}$ corresponding to the total protein concentration, and the second equality reflects our assumption that the "ribosome" reaction r is the only one producing proteins (with no reaction consuming them), so that $M_{j}^{\mathrm{a}} = 0$ for $j \neq \mathrm{r}$. Equating the right hand sides of the previous two Eqs 14 and 15 and solving for $\mu$ (with $b^{\mathrm{a}} \neq 0 \Rightarrow \mathbf{f}^{\top}\boldsymbol{\tau}(\mathbf{f}) \neq 0$), we get the *growth function*

$$\mu(\mathbf{f}) = \frac{M_{\mathrm{r}}^{\mathrm{a}}f^{\mathrm{r}}}{\mathbf{f}^{\top}\boldsymbol{\tau}(\mathbf{f})} \quad . \tag{16}$$

Thus, the growth rate becomes an explicit function of only the flux fractions $\mathbf{f}$. $\mu$ still depends on the fixed parameters $\rho$, $\mathbf{M}$, and $\mathbf{x}$ through the functions $\boldsymbol{\tau} = \boldsymbol{\tau}(\rho\,\mathbf{M}\,\mathbf{f}, \mathbf{x})$. Note that if fluxes $\mathbf{v}$ were used instead of the flux fractions $\mathbf{f}$, then $\boldsymbol{\tau}(\mathbf{c}, \mathbf{x}) = \boldsymbol{\tau}(\mathbf{M}\,\mathbf{v}/\mu, \mathbf{x})$, which would cause a recursion issue when defining the growth rate as a function of $\mathbf{v}$ and $\boldsymbol{\tau}$ following the same procedure [7, 23]. In that case, one is forced to account for $\mathbf{c}$ as separate variables, thereby increasing the dimensionality of the problem. The same recursion issue occurs when formulating the problem in terms of protein concentrations $\mathbf{p}$.

From now on, we will consider $\mathbf{b}$ (Eq 11) and $\boldsymbol{\tau}$ (Eq 12) as functions of $\mathbf{f}$, and treat $\mathbf{f}$ as the only free variables. After writing the growth rate $\mu$ as a function of $\mathbf{f}$, we now do the same thing for our remaining constraints, so now we have much fewer variables and constraints.

In the scaled variables, the density constraint (Eq 6) is reduced to

$$\sum_i b^i = 1 \quad .$$

(17)

Using Eq 11, we can rewrite this constraint in terms of flux fractions $\mathbf{f}$. We see that in balanced growth, the density constraint (Eq 17) is equivalent to a flux balance on the cell surface,

$$1 = \gamma^\top \mathbf{f} = \sum_s \gamma_s f^s \quad ,$$

(18)

where the second equality comes from Eq 2: only the columns $s$ sum up to non-zero values $\gamma_s$, so only transport fluxes $f^s$ are limited by this constraint. The nature of this constraint as a global mass balance becomes more evident if we multiply the whole expression by $\mu\rho$: the net mass uptake $\sum_s \gamma_s v^s$ going through the cell surface must equal the rate of biomass production $\mu\rho$.

Any solution to the growth function (Eq 16) automatically respects internal mass conservation, protein density and the kinetic constraints: for any given vector $\mathbf{f}$, $\mu(\mathbf{f})$ returns the unique growth rate satisfying these constraints (which also depend on $\rho$ through $\boldsymbol{\tau} = \boldsymbol{\tau}(\rho\,\mathbf{M}\,\mathbf{f}, \mathbf{x})$). The flux balance at the cell surface is enforced separately by Eq 18 on transporters, making these fundamentally different from enzymatic and ribosome reactions. In particular, for a model with only one transporter $s$, Eq 18 already determines the scaled uptake rate $f_s = (\gamma_s)^{-1}$. With two transport fluxes, one flux is uniquely determined by the other; a simple example would be a model that only has transporters for glucose uptake and $CO_2$ excretion (see example model "C" in S1 Text). More generally, Eq 18 can be used to uniquely determine one transport flux fraction in terms of the others, reducing the number of variables by one. For clarity of presentation, however, we will keep Eq 18 as a separate constraint and not eliminate any variable, until the introduction of growth control coefficients in the corresponding section.

Finally, writing the non-negativity constraints on proteins and reactant concentrations in terms of $\mathbf{f}$ results in the following element-wise inequalities on the corresponding vectors

$$\mathbf{f} \odot \boldsymbol{\tau}(\mathbf{f}) = \frac{\mathbf{p}}{\mu\rho} \geq \mathbf{0} \quad ,$$

(19)

$$\mathbf{M}\,\mathbf{f} = \mathbf{b} = \frac{\mathbf{c}}{\rho} \geq \mathbf{0} \quad .$$

(20)

We are now in the position to provide a concise formulation of growth rate optimization in terms of flux fractions $\mathbf{f}$. Combining Eqs 16, 18, 19, 20, the optimal growth problem for a given environment $\mathbf{x}$ becomes

$$\underset{\mathbf{f}\,\in\,\mathbb{R}^N}{\text{maximize}} \qquad \mu(\mathbf{f}) = \frac{M_r^a f^r}{\mathbf{f}^\top \boldsymbol{\tau}(\mathbf{f})}$$

subject to :

$$\gamma^\top \mathbf{f} = 1$$

$$\mathbf{f} \odot \boldsymbol{\tau}(\mathbf{f}) \geq \mathbf{0}$$

$$\mathbf{M}\,\mathbf{f} \geq \mathbf{0} \quad ,$$

(21)

where $\mathbf{f}$ is a vector containing a real-valued flux fraction for each reaction ($\mathbf{f} \in \mathbb{R}^N$, with $N :=$ number of columns in $\mathbf{M}$), and the turnover times $\boldsymbol{\tau} = \boldsymbol{\tau}(\rho\,\mathbf{M}\,\mathbf{f}, \mathbf{x})$ are functions that depend on

**f** and on the parameters $\rho$, **M**, **x**. In the following discussion, we assume that all $\tau^j$ are different from zero, which simply means there is no reaction with infinite turnover rate (this relationship can be visualized in Fig 1C). We note that the direction of reactions (i.e. the sign of each $f^j$) is not enforced here such as in some other methods, but instead emerges as a result of the optimization (21); because of the constraint 19, a non-zero $f^j$ should always have the same sign as $\tau^j$, which is in turn a thermodynamic property of reaction $j$ determined by its kinetic parameters and the relevant reactant concentrations [31]. If the rate laws have a general functional form, these functions will be parameterised by the set of kinetic parameters $\mathbb{K}$. After solving this optimization problem, all original cellular variables (unscaled fluxes as well as unscaled metabolite and protein concentrations) can be easily reconstructed from **f**. In the following, we will refer to $\boldsymbol{\pi}$ as the vector of parameters that define the optimization problem, which includes **M**, $\rho$, and **x**, as well as the elements of $\mathbb{K}$. The parameters in $\boldsymbol{\pi}$ are considered fixed until the section "Growth Control and Adaptation", where we study the sensitivity of optimal growth to marginal changes in the components of $\boldsymbol{\pi}$.

Table 1 lists all symbols used below.

## Growth analysis

Next, we utilize the problem reformulation in terms of flux fractions **f** to derive general necessary conditions of OGSs, valid for any GBA model—including those with redundant pathways. First, for each reaction, we will derive explicit expressions for *shadow prices* in the optimal state; in constrained optimization in economics, the shadow price is the change, per infinitesimal unit of the constraint, in the optimal value of the objective function of an optimization problem obtained by relaxing the constraint. This term has been applied also to biological systems in the context of constraint-based optimization. [32]. We then derive equations for the state variables **f** themselves, which must hold in any optimal state. This development constitutes a generalization of our previous analytical approach to GBA [4], which was restricted to models with matrices **M** of full column rank. The latter condition is not generally satisfied by realistic cellular models, as many cellular biochemical reactions are structurally redundant, i.e., their columns in **M** are linearly dependent on other columns. OGSs always have non-redundant active reactions (i.e., the *active* **M** has full column rank) [4, 21, 22], but this optimal set of active reactions is generally not known *a priori*. In contrast, the following analysis in terms of flux fractions is valid for any **M** of arbitrary size and rank.

For the following, we emphasize that the state of our system is completely determined by scaled fluxes $f^j$, which serve as independent variables. All other variables are fully dependent on them: the unscaled fluxes **v**, the scaled and unscaled concentrations **b**, **c**, and **p**, the reaction times $\boldsymbol{\tau}$, and the growth rate $\mu$.

All following analyses benefit from knowing the system's sensitivity to small changes of each of the independent variables $f^j$. The partial derivatives of the system's properties $\mathbf{c}(\mathbf{f})$, $\mathbf{v}(\mathbf{f})$, $\mathbf{b}(\mathbf{f})$, $\boldsymbol{\tau}(\mathbf{f})$, $\mathbf{p}(\mathbf{f})$, and $\mu(\mathbf{f})$ with respect to each $f^j$ provide explicit expressions for sensitivity coefficients similar to the ones introduced in Metabolic Value Theory [32], based on the original concepts of Metabolic Control Analysis (MCA) [9, 10]. A unique feature of the present treatment arises from the system of equations in Eq 11, which determines the linear dependence of biomass fractions **b** on **f**, so that the partial derivative of $b^i$ with respect to $f^j$ is given simply as

$$\frac{\partial b^i}{\partial f^j} = M^i_j \quad . \tag{22}$$

Via the chain rule of differentiation, this expression also determines the partial derivatives with respect to $f^j$ for any functions of $b^i$. A case of particular interest in the following

**Table 1. Symbols used.**

| Symbol | Description (units) |
|---|---|
| $A$ | growth adaptation coefficient |
| $\mathbf{b}$ | biomass fraction vector |
| $\mathbf{c}$ | reactant concentration vector ([mass][volume]$^{-1}$) |
| $\mathbf{C}$ | control coefficient matrix |
| $\mathbf{f}$ | flux fraction vector |
| $\mathbf{E}$ | indirect sensitivity matrix ([time]) |
| $\mathbb{K}$ | set of kinetic parameters (various units) |
| $\mathcal{L}$ | Lagrangian ([time]$^{-1}$) |
| $\mathbf{M}$ | mass fraction matrix |
| $N$ | number of reactions (= number of columns in $\mathbf{M}$) |
| $\mathbf{p}$ | protein concentration vector ([mass][volume]$^{-1}$) |
| $\mathbf{S}$ | stoichiometric matrix ([mol]) |
| $\mathbf{v}$ | flux vector ([mass][volume]$^{-1}$[time]$^{-1}$) |
| $\mathbf{x}$ | external reactant concentration vector ([mass][volume]$^{-1}$) |
| $\boldsymbol{\gamma}$ | vector with column sums of $\mathbf{M}$ |
| $\Gamma$ | growth control coefficient ([time]$^{-1}$) |
| $\boldsymbol{\varepsilon}$ | direct sensitivity matrix ([time]) |
| $\boldsymbol{\phi}$ | proteome mass fraction vector |
| $\theta$ | KKT multiplier for the protein non-negativity constraint ([time]$^{-2}$) |
| $\lambda$ | KKT multiplier for the density constraint ([time]$^{-1}$) |
| $\mu$ | growth rate ([time]$^{-1}$) |
| $\boldsymbol{\pi}$ | parameter vector (various units) |
| $\rho$ | mass density ([mass][volume]$^{-1}$) |
| $\boldsymbol{\tau}$ | turnover time vector ([time]) |
| **Index** | **Description** |
| $a$ | all proteins |
| $e$ | enzymatic reactions |
| $m$ | metabolites |
| $i$ | internal reactants (including $m$ and a) |
| $r$ | ribosome reaction |
| $s$ | surface reactions (i.e., transport reactions) |
| $j$ | reactions (including $s$, $e$, r) |
| $l$ | reactions (including $s$, $e$, r) |

discussions is the vector of turnover times $\boldsymbol{\tau} = \boldsymbol{\tau}(\mathbf{c}, \mathbf{x}) = \boldsymbol{\tau}(\rho\, \mathbf{b}, \mathbf{x}) = \boldsymbol{\tau}(\rho\, \mathbf{M}\, \mathbf{f}, \mathbf{x})$. We first define the (direct) *time elasticities* (*elasticities* in short), the sensitivity of each turnover time $\tau^l(\mathbf{c}, \mathbf{x}) = \tau^l(\rho\, \mathbf{b}, \mathbf{x})$ with respect to each biomass fraction $b^i$, as

$$\varepsilon_i^l := \frac{\partial \tau^l}{\partial b^i} = \frac{\partial \tau^l}{\partial c^i}\frac{\partial c^i}{\partial b^i} = \frac{\partial \tau^l}{\partial c^i}\rho \quad , \tag{23}$$

where we used the chain rule of differentiation in the first equality and Eq 9 in the second. We then use the direct elasticities $\varepsilon_i^l$ to express the sensitivity of $\tau^l$ to a change in a flux fraction $f^j$, defined as the *indirect time elasticity* matrix $\mathbf{E}$ (or *indirect elasticity* in short), with entries

$$E_j^l := \frac{\partial \tau^l}{\partial f^j} = \sum_i \frac{\partial \tau^l}{\partial c^i}\frac{\partial c^i}{\partial b^i}\frac{\partial b^i}{\partial f^j} = \sum_i \varepsilon_i^l M_j^i \quad , \tag{24}$$

where we used Eq 22 in the last equality. In the following discussion, we assume that the kinetic rate laws do not depend on the total protein concentration $c^a$, meaning $\varepsilon^l_a = \partial \tau^l / \partial c^a = 0$ for all reactions $l$. That would be different if, for example, one accounts for the macromolecular crowding effects via kinetic rate laws [33]. The indirect elasticities $\mathbf{E}$ and direct elasticities $\boldsymbol{\varepsilon}$ share some resemblance with the Jacobian and elasticity matrices defined in Metabolic Value Theory and MCA, although we do not intend to explore the exact relationships in this work. For an example of direct and indirect elasticities, where $\boldsymbol{\tau}$ follows a simple Michaelis-Menten rate law, see S1 Text.

In the remainder of this paper, we will explore three complementary types of analyses of GBA systems. First, in the growth optimality section we will state the analytical conditions necessary for an optimal state $\mathbf{f}^*$. Second, in the growth economy section we will calculate the sensitivity of a (not necessarily optimal) growth rate $\mu$ to small changes in each $\mathbf{f}$, which we interpret in economic terms as marginal values of reactions. Third, in the growth control and adaptation section we will estimate the sensitivity of the optimal growth rate $\mu^*$ to small changes in the previously fixed parameters $\boldsymbol{\pi}$. In each of these analyses, the sensitivity measures captured by $\mathbf{E}$ will appear naturally in the results.

**Growth optimality.** We next calculate the necessary analytical conditions for the optimal growth state (OGS). This calculation extends our previous analytical approach, which was restricted to GBA models with matrices $\mathbf{M}$ of full column rank [4], to general GBA models with arbitrary matrices $\mathbf{M}$, facilitated by the reformulation of the GBA problem in terms of flux fractions $\mathbf{f}$. We approach this problem by studying the Karush-Kuhn-Tucker (KKT) conditions [34, 35], which generalize the method of Lagrange multipliers by also accounting for inequality constraints, here present due to the non-negativity of concentrations. To simplify the presentation in this section, we here account explicitly only for the non-negativity of protein concentrations, but not for the non-negativity of metabolite concentrations. Under the reasonable assumption that metabolites with zero concentration do not participate in any active reactions, the resulting necessary conditions are also necessary when accounting for this latter constraint; the full calculations can be found in S1 Text.

We define the Lagrangian $\mathcal{L}(\mathbf{f}, \lambda, \boldsymbol{\theta})$ for a given GBA model $(\mathbf{M}, \boldsymbol{\tau}, \rho)$ and external concentrations $\mathbf{x}$ as

$$\mathcal{L}(\mathbf{f}, \lambda, \boldsymbol{\theta}) := \mu(\mathbf{f}) + \lambda(\gamma^\top \mathbf{f} - 1) + \boldsymbol{\theta}^\top \mathbf{f} \odot \boldsymbol{\tau}(\mathbf{f}) \quad , \tag{25}$$

The *KKT multipliers* $\lambda$, $\boldsymbol{\theta}$ are auxiliary variables used to find the optimal state, but also encode important economic and control information about the system at optimality, as we will see later. $\lambda$ relates to the equality constraint enforcing the fixed cell density, connected to $\mathbf{f}$ via the flux balance at the surface (Eq 18); $\boldsymbol{\theta}$ relate to the inequality constraint enforcing the non-negativity of proteins (Eq 19). Solving the KKT conditions (see Methods for details), we get the *balance equations* determining the necessary condition for each reaction $j$ at optimal growth:

$$(\partial_j \mu + \lambda \gamma_j) f_j = 0 \quad , \tag{26}$$

where $\partial_j \mu := \partial \mu / \partial f^j$ indicates the partial derivative of $\mu$ with respect to $f^j$, calculated from Eq 16 as

$$\partial_j \mu = \frac{\mu}{b^a} \left( M^a_j - \mu \tau_j - \mu \mathbf{f}^\top \mathbf{E}_j \right) \quad , \tag{27}$$

with $\mathbf{E}_j$ representing the column $j$ of $\mathbf{E}$, and $\lambda$ is the optimal value of the density constraint

multiplier

$$\lambda = \frac{\mu^2}{b^{\mathrm{a}}}\, \mathbf{f}^\top\, \mathbf{E}\, \mathbf{f} \tag{28}$$

(see Methods for the detailed calculations). When we further consider that only transporters have a nonzero column sum $\gamma_j$ (Eq 2), we get an equivalent expression for the optimal $\lambda$ that highlights its particular dependence on the reactions directly connected to the transport reactions (see S1 Text).

Eq 26 is a necessary property of the global optimum of (21), but it could be in principle also satisfied by other local stationary points of the constrained growth rate defined by Eq. 21, which exist whenever the optimization (21) is non-convex. The possible non-uniqueness of solutions for Eq 26 is however not of our concern here, since in this study we are focusing only on the analytical properties of the global optimum, not on methods to calculate it.

Eq 26 generalizes the necessary analytical conditions we found before [4] for the optimal states of GBA models with full column rank matrix $\mathbf{M}$; in that case, the conditions could only be applied to arbitrary models if one had prior knowledge of what reactions are active at optimality, effectively reducing $\mathbf{M}$ to an "active" matrix of interest (which is guaranteed to be of full column rank [4]). Here, no prior knowledge of active reactions is required. Instead, Eq 26 provides the very condition determining whether each reaction is active at optimality: a reaction with nonzero flux $f_j$ requires that the corresponding term in parentheses (i.e., the corresponding $\theta_j$, see Methods) is equal to zero. Conversely, if the term in parentheses is different from zero ($\theta_j \neq 0$), then the reaction cannot carry flux at optimality ($f_j = 0$). In particular, the ribosome evidently needs to be active for balanced cellular growth, as proteins are required as catalysts; thus, $\theta_{\mathrm{r}} = 0$ must always hold in optimal states. The KKT multipliers $\boldsymbol{\theta}$ are the *shadow prices* [32] of each $\tau^j f^j = p^j/(\mu\rho)$, which has unit of time, and can be understood as the fraction $p^j/\rho$ of the total growth time $1/\mu$ which is allocated to produce the protein $j$ in the biomass.

We may also express Eq 26 for each reaction $j$ in the usual, unscaled variables $\mathbf{v}$ (fluxes), and $\mathbf{c}$ (reactant concentrations, including metabolites and total protein), by using Eqs 4, 9, 10 and 11 (see S1 Text)

$$\left( M_j^{\mathrm{a}} - \mu\,\tau_j - \mathbf{v}^\top \boldsymbol{\varepsilon}\, \mathbf{M}_j + \mathbf{v}^\top \boldsymbol{\varepsilon}\, \frac{\mathbf{c}}{\rho}\, \gamma_j \right) v_j = 0 \quad , \tag{29}$$

where $\mathbf{M}_j$ indicates the column $j$ of $\mathbf{M}$. We now continue our analysis in terms of the flux fractions $\mathbf{f}$, since these are the variables of the optimization problem (Eq. 21). However, we keep in mind that the same change of variables to $\mathbf{p}$, $\mathbf{v}$, $\mathbf{c}$ is possible in all the following equations, as done for Eq 29.

**Growth economy.** As growth rate is closely related to fitness [2], it makes sense to view growth rate as the primary value of the cellular economy. In this subsection, we will thus explore the economy of balanced cellular growth, by asking how a small change in the state variables $f^j$ affects the growth rate $\mu$ of any optimal or non-optimal state. Below, we will see that the necessary conditions of optimal growth specify that the marginal costs and benefits of each flux must be perfectly balanced.

We define the *marginal value* of flux $j$ as the partial derivative $\partial_j \mu$, which quantifies the marginal gain in growth rate resulting from a small increase in $f^j$. From Eq 27, we see that the marginal value can be expressed naturally as a multiple of the growth rate per mass fraction of protein in biomass, $\mu/b^{\mathrm{a}}$. As we will see next, the corresponding adimensional factor — the term in parentheses in Eq 27—quantifies different types of costs (when negative) and benefits

(when positive) of reaction $j$ in terms of its influence on protein allocation,

$$\frac{b^{\mathrm{a}}}{\mu}\,\partial_j \mu = M_j^{\mathrm{a}} - \mu\,\tau_j - \mu\,\mathbf{f}^{\top}\mathbf{E}_j \quad . \tag{30}$$

The first summand quantifies how a marginal increase in $f^j$ increases the total protein fraction in the cell density $b^{\mathrm{a}} = c^{\mathrm{a}}/\rho$ (see Eq 5),

$$M_j^{\mathrm{a}} = \frac{\partial b^{\mathrm{a}}}{\partial f^j} \quad . \tag{31}$$

We name this contribution to the normalized marginal value the *marginal protein production*. As we assume that the ribosome reaction is the only reaction that consumes or produces protein, this reaction ($j = \mathrm{r}$) is the only one with a nonzero (and positive) marginal production benefit.

To interpret the remaining summands, we remember that an individual protein's mass fraction in the cellular density can be expressed as $p^l/\rho = \tau^l v^l/\rho = \mu\tau^l f^l$. The last two terms in Eq 30 quantify the combined decrease of individual protein fractions in cellular density ($p^l/\rho$) caused by a marginal increase in $f^j$ at fixed $\mu$,

$$-\mu\,\tau_j - \mu\,\mathbf{f}^{\top}\mathbf{E}_j = -\left(\frac{\partial(\mu\,\mathbf{f}^{\top}\boldsymbol{\tau})}{\partial f^j}\right)_{\mu} = -\sum_l \left(\frac{\partial(p^l/\rho)}{\partial f^j}\right)_{\mu} \quad . \tag{32}$$

Here, the first summand quantifies the change in $p^l/\rho$ at fixed turnover times, which is evidently non-zero only for the enzyme catalyzing the perturbed flux $j$ itself. We name this term, $-\mu\tau_j$, the *marginal (protein) investment* into $j$. The final summand quantifies the local change of the individual protein concentrations that must occur to compensate the changes in the turnover times (quantified by the indirect elasticity $\mathbf{E}$), themselves caused by changes in metabolite concentrations forced due to flux balance. We name it the *marginal (protein) opportunity* of $j$, as it is related to opportunity costs and benefits in economics. For the typical case of reactions running in the forward direction ($f^j > 0$), $\tau^j$ is positive, and thus the marginal investment into $j$ is negative, representing a cost. If all fluxes are non-negative, beneficial decreases in turnover times correspond to negative $E$, resulting in positive marginal opportunity (i.e., marginal opportunity benefit).

We can now summarize our insights about cellular economy, in particular about changes in the growth rate $\mu$ in response to changes in a flux $f^j$. The first and second terms in Eq 30 are simple, direct consequences of the flux change: the marginal production benefit, an increase in protein production if $f^j$ is the ribosome flux; and the marginal investment, an increase in the protein concentration required to sustain an increased $f^j$. The third term in Eq 30, the marginal opportunity, is more interesting, though equally easy to understand. As a simple consequence of mass conservation (Eq 11), a change in $f^j$ while keeping all other fluxes fixed must result in changes in the concentrations of all reactants consumed or produced in the corresponding reaction. These concentration changes modify the turnover times $\tau_l(\mathbf{c})$ of all reactions $l$ whose kinetics depend on them, either because they are directly connected to those reactants or because they act as inhibitors or activators; see Fig 3 for an example. Keeping the corresponding fluxes $f^l$ constant requires matching changes in the concentrations $p_l$ of the catalyzing proteins (Eq 4). This total amount of "protein saved" due to a change in $f_j$ is quantified by $-\mu\sum_l f_l E_j^l$.

The above results confirm the often postulated central role of proteins in the cellular economy [31, 36, 37]. While the measure of cellular economic value may be the growth rate itself, protein concentrations constitute the general currency in which we can express the

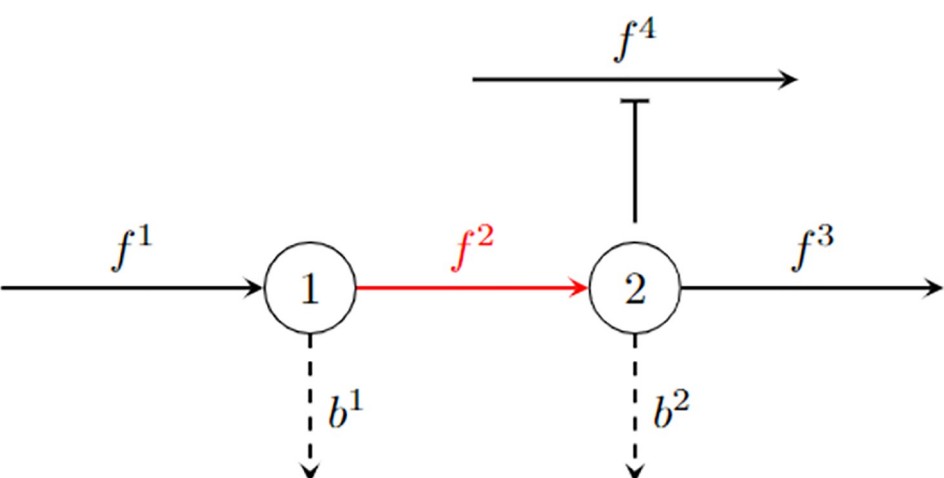

**Fig 3. The dependence of marginal opportunity on the reaction neighborhood.** The figure shows a simple example of a reaction ($j = 2$, red) that is directly connected to two metabolites ($m = 1, 2$) and thereby to two other reactions $j = 1, 3$. Reaction $j = 2$ is also connected indirectly to reaction $j = 4$ by inhibiting it through metabolite $m = 2$ (indicated by the blunt arrow $\top$). The marginal opportunity of reaction $j = 2$ is
$-\mu \mathbf{f}^\top \mathbf{E}_2 = -\mu \left( f_1 \frac{\partial \tau^1}{\partial b^1} M_2^1 + f_2 \frac{\partial \tau^2}{\partial b^1} M_2^1 + f_2 \frac{\partial \tau^2}{\partial b^2} M_2^2 + f_3 \frac{\partial \tau^3}{\partial b^2} M_2^2 + f_4 \frac{\partial \tau^4}{\partial b^2} M_2^2 \right)$. It quantifies how a marginal change in $f^2$ while keeping all other $f^j$ fixed, causes (i) an inevitable change in $b^1, b^2$ due to the flux balance (Eq 11); which by consequence causes (ii) an inevitable change in $\tau^1, \tau^2, \tau^3, \tau^4$, as these are functions of $b^1, b^2$; which finally causes (iii) an inevitable change in $p^1, p^2, p^3, p^4$ due to the kinetic constraints (Eq 4) at fixed $v^1, v^2, v^3, v^4$ (determined by the fixed flux fractions and growth rate (Eq 10)). The example also shows how the information about mass conservation and reaction kinetics is completely built into the definition of the growth function (Eq 16).

contributions of cellular subsystems. We can highlight this central economic role of proteins further by relating the marginal values $\partial_j \mu$—changes in growth rate in response to flux changes—to changes in the allocation of proteome fractions $\phi^l := p^l/c^a$:

$$\partial_j \mu = \frac{\mu}{b^a} \left( M_j^a - \mu \tau_j - \mu \mathbf{f}^\top \mathbf{E}_j \right) = -\mu \sum_l \left( \frac{\partial \phi^l}{\partial f^j} \right)_\mu \quad . \tag{33}$$

The second equality follows directly from taking the derivative of $\sum_l \phi^l = \mu \rho \, \mathbf{f}^\top \boldsymbol{\tau}/\rho \, \mathbf{M}^a \, \mathbf{f}$ with respect to $f^j$, at constant $\mu$.

We can now look at the balance equations at optimal growth from an economic perspective. For the ribosome and active enzymatic reactions, a zero marginal value $\partial_j \mu = 0$ also means a zero shadow price, $\theta_j = 0$ (Eqs 54 and 55)—so the reaction is optimal, and growth cannot be accelerated by increasing or decreasing $f^j$ by a small amount. This insight provides an intuitive interpretation for the balance equations for the ribosome (Eq 57) and for all active, internal enzymatic reactions (Eq 58). An exception are only the transporters. In contrast to all other flux fractions, their shadow price (Eq 56) depends both on their marginal value and on their *marginal biomass production*, $\mu \, \mathbf{f}^\top \mathbf{E} \, \mathbf{f} \, \gamma_s$ (a cost when negative and a benefit when positive).

For active enzymes with zero marginal value—and thus for all active enzymes at optimality (Eq 58)—Eq 27 simplifies to

$$\tau_e + \mathbf{f}^\top \mathbf{E}_e = 0 \quad . \tag{34}$$

This simple relationship shows that at optimality, the marginal investment into $e$ should perfectly balance its marginal opportunity. As the last equation involves only the

neighborhood of $e$ (defined as all reactions $l$ such that $E_e^l \neq 0$), we can study such relationships at optimality locally, without full knowledge about the entire reaction network. We thus do not need the entire matrix **M** or complete knowledge of parameterized turnover time functions in the vector $\boldsymbol{\tau}$.

In the preceding subsection, we studied how any optimal or non-optimal growth rate $\mu$ is sensitive to marginal changes in one of the flux fractions, resulting in an economic understanding of marginal flux values in terms of their relationship with protein allocation. We next reinterpret some of these results from the perspective of control theory, and turn to a complementary problem that focuses on the sensitivity of the optimal growth rate to changes in the model parameters and external concentrations.

**Growth control and adaptation.** We are first interested in the total control that each $f^j$ has on the (optimal or non-optimal) growth rate $\mu$, accounting also for the density constraint limitation. In order to do that, we choose one active transport reaction $s'$ and express its corresponding $f^{s'} \neq 0$ as a function of the other fluxes via the density constraint (Eq 18),

$$f_{s'} = \frac{1}{\gamma_{s'}} \left( 1 - \sum_l \gamma_l f^l \right) \quad , \tag{35}$$

where $l \neq s'$ sums over all other reactions. Thus,

$$\frac{\partial f^{s'}}{\partial f^j} = -\frac{\gamma_j}{\gamma_{s'}} \quad , \tag{36}$$

which is non-zero only if $j$ is also a transport reaction (so $\gamma_j \neq 0$). We now define the *Growth Control Coefficients* $\Gamma_j$ as

$$\Gamma_j := \partial_j \mu - \partial_{s'} \mu \, \frac{\gamma_j}{\gamma_{s'}} \quad , \tag{37}$$

where the first term quantifies the growth change caused by $f^j$ itself, and the second term quantifies the growth change caused by a change in $f^{s'}$, itself changed due to the changed $f^j$ and the density constraint. Note that for the ribosome and enzymatic reactions, their growth control coefficient is simply their marginal value, since $\gamma_r = \gamma_e = 0$. For models with only one transport reaction $s$, $\Gamma_s = 0$, since $f_s = (\gamma_s)^{-1}$ is fixed by the density constraint and cannot be changed. Conveniently, this is also captured by Eq 37. If $s'$ is optimal, $\theta_{s'} = 0$, and Eq 26 determines $\partial_{s'} \mu = -\lambda \gamma_{s'}$, so in that case

$$\Gamma_j = \partial_j \mu + \lambda \, \gamma_j \quad , \tag{38}$$

and the balance equation Eq 26 is thus equivalent to

$$\Gamma_j f_j = 0 \quad . \tag{39}$$

We may also see the optimal condition for enzymes (Eq 34) in terms of protein concentrations, by multiplying it element-wise with $\nu_e$ (so it is also valid now for inactive enzymes),

$$p_e = \sum_l p_l \, C_e^l \quad , \tag{40}$$

where we defined

$$C_e^l := \frac{f_e}{\nu^l} \left( \frac{\partial \nu^l}{\partial f^e} \right)_{p^l} = \frac{f_e}{\nu^l} \left( \frac{\partial (p^l / \tau^l)}{\partial f^e} \right)_{p^l} = -\frac{f_e}{\tau^l} \frac{\partial \tau^l}{\partial f^e} \quad , \tag{41}$$

$$\boldsymbol{\pi} \xrightarrow{\text{optimization (21)}} \mathbf{f}^* \xrightarrow{Eqs.\ 4,9,10,16} \mathbf{p}^*, \mathbf{c}^*, \mathbf{v}^*, \mu^*$$

**Fig 4. The parameters $\pi$ and their control on the optimal cellular state f\*.**

via Eq 4 and using partial derivatives at fixed $p^l$. $C_e^l$ can be seen as (scaled) *control coefficients* (CC), analogous to (scaled) control coefficients in MCA [9, 10]. This result is analogous to how enzyme concentrations and their respective CC relate at optimal fluxes constrained by a fixed total enzyme concentration [38] (see S1 Text for a detailed discussion). For an example of control coefficients where $\tau$ follows a simple Michaelis-Menten rate law, see S1 Text.

We now explore the sensitivity of the optimal growth rate to changes in one parameter $\pi$ in the vector $\boldsymbol{\pi}$. The growth problem (Eq 21) is constrained by the parameters $\boldsymbol{\pi}$, including the arguments necessary to determine the turnover times $\tau$ at given $\mathbf{f}$. This means that any marginal change in one of the parameters $\boldsymbol{\pi}$ would lead to changes in the solution $\mathbf{f}^*$ of the optimization (Eq 21). In this sense, the parameters $\boldsymbol{\pi}$ can be understood as control variables, while the corresponding optimal state $\mathbf{f}^*$, and its functions $\mu^* = \mu(\mathbf{f}^*)$, $\mathbf{v}^* = \mathbf{v}(\mathbf{f}^*)$, $\mathbf{p}^* = \mathbf{p}(\mathbf{f}^*)$, and $\mathbf{c}^* = \mathbf{c}(\mathbf{f}^*)$ are the response variables. Fig 4 summarizes these relationships.

Because growth rate is closely related to fitness, we are also particularly interested in how marginal changes in one of the previously fixed parameters $\boldsymbol{\pi}$ affect the *optimal* growth rate $\mu^*$ [39]. We can estimate this effect directly via the envelope theorem [4, 40], by effectively considering the *optimal* state $\mathbf{f}^*$ as fixed and treating the parameters $\boldsymbol{\pi}$ as the new independent variables, making it unnecessary to calculate the new optimal state after the parameter change. To do that, we first simplify the problem by assuming that these marginal changes have no effect on which reactions are active, so we simplify the Lagrangian (Eq 25) by ignoring the inequality constraints; note that in this case only the objective function $\mu$ can be influenced by parameter changes, since the density constraint only depends on $\mathbf{M}$, whose entries cannot be changed continuously. Second, we can think about the optimal growth rate $\mu^*$ as a function of the parameters $\mu^*(\boldsymbol{\pi}) := \mathcal{L}(\mathbf{f}^*(\boldsymbol{\pi}), \boldsymbol{\lambda}^*(\boldsymbol{\pi}), \boldsymbol{\pi})$, so the total change $\mathrm{d}\mu^*/\mathrm{d}\pi$ induced by a marginal change in a parameter $\pi$ can be calculated via the chain rule

$$\frac{\mathrm{d}\mu^*}{\mathrm{d}\pi} = \frac{\partial \mathcal{L}}{\partial f_j} \frac{\mathrm{d} f_j^*}{\mathrm{d}\pi} + \frac{\partial \mathcal{L}}{\partial \lambda} \frac{\mathrm{d}\lambda_\rho^*}{\mathrm{d}\pi} + \frac{\partial \mathcal{L}}{\partial \pi} = \frac{\partial \mathcal{L}}{\partial \pi} \quad , \tag{42}$$

where the last equality comes from $\partial \mathcal{L}/\partial f^j = \partial \mathcal{L}/\partial \lambda = 0$ according to the stationarity (Eq 49) and primal feasibility (Eq 18) at an optimal state.

We now define *growth adaptation coefficients A* as the relative change in the *optimal* growth rate $\mu^*$ in response to a small, relative change in one control variable $\pi$

$$A_\pi := \frac{\pi}{\mu^*} \frac{\mathrm{d}\mu^*}{\mathrm{d}\pi} = \frac{\pi}{\mu(\mathbf{f})} \frac{\partial \mathcal{L}}{\partial \pi} \quad , \tag{43}$$

where here and in the rest of this section $\mathbf{f}$ is to be understood as the *optimal* state before the change in the parameter $\pi$. Note that in the following discussion, the parameters $\pi$ of interest only influence $\mathcal{L}$ via the objective function $\mu$, so the partial derivatives $\partial \mathcal{L}/\partial \pi$ are simply evaluated as $\partial \mu/\partial \pi$ at fixed $\mathbf{f}$.

For direct changes in the turnover times $\tau^j$ (e.g., through changing the corresponding $1/k_{\text{cat}}^j$), the growth adaptation coefficient is calculated by evaluating the growth function $\mu$ and

its partial derivative at fixed **f**,

$$A_{\tau^j} := \frac{\tau_j}{\mu^*} \frac{d\mu^*}{d\tau^j} = \frac{\tau_j}{\mu} \frac{\partial \mu}{\partial \tau^j} = -\frac{\mu \rho f_j \tau_j}{c_a} = -\phi_j \quad , \tag{44}$$

where we effectively treated $\tau^j$ as a variable in the growth equation Eq 16, and $\phi_j = p_j/c_a$ is the optimal proteome fraction allocated to reaction $j$ before the change in $\tau^j$. This result is consistent with the observation that drugs targeting the most highly expressed catalysts, such as the ribosome, have the strongest effects on cellular growth rates [5, 41].

For changes in some external parameter such as a concentration $x$, the growth adaptation coefficient is again calculated by evaluating the growth function $\mu$ and its partial derivative at fixed **f**, and using the chain rule of differentiation we obtain

$$A_x := \frac{x}{\mu^*} \frac{d\mu^*}{dx} = \frac{x}{\mu} \sum_s \frac{\partial \mu}{\partial \tau^s} \frac{\partial \tau^s}{\partial x} = -\sum_s \phi_s \frac{x}{\tau^s} \frac{\partial \tau^s}{\partial x} \quad , \tag{45}$$

where we have a summation over $s$ (only transporters $s$ have kinetic rate laws depending on external concentrations). According to Eq 45, the growth adaptation coefficient of an external concentration $x$ is simply the sum over the "scaled elasticities" $\dfrac{x}{\tau^s} \dfrac{\partial \tau^s}{\partial x}$ of the transporters of $x$, weighted by the optimal proteome fractions $\phi_s$ allocated to each $s$ before the change in $x$. This result gives an explicit quantitative estimation on which external concentrations should be changed in order to cause the most change in the optimal growth rate. This equation may hence provide a useful tool for improving the growth media environment for industrial cell cultures, and for quantifying the effect of drugs aimed at decreasing the growth of pathogens and cancer cells. If the turnover times $\tau$ depend explicitly on other external parameters, such as pH and temperature, growth adaptation coefficients can be calculated and interpreted exactly as in Eq 45.

The growth adaptation coefficient with respect to the mass density $\rho$, assuming it affects turnover times $\tau$ only through reactant concentrations **c**, reads

$$A_\rho := \frac{\rho}{\mu^*} \frac{d\mu^*}{d\rho} = \frac{\rho}{\mu} \sum_{l,i} \frac{\partial \mu}{\partial \tau^l} \frac{\partial \tau^l}{\partial c^i} \frac{\partial c^i}{\partial \rho} = -\frac{\rho}{\mu} \sum_{l,i,j} \left( \frac{\mu^2}{b^a} f_l \frac{\partial \tau^l}{\partial c^i} M_j^i f^j \right) = -\frac{\mu}{b^a} \mathbf{f}^\top \mathbf{E} \mathbf{f} \quad , \tag{46}$$

where $\partial c^i / \partial \rho = b^i = \sum_j M_j^i f^j$ according to Eqs 9 and 11, and the last equality comes from the definition of the indirect elasticity (Eq 24). From this expression and $\lambda$ in Eq 28, we see that $-\lambda = \mu A_\rho$; at optimality, the negative KKT multiplier for the density constraint, $-\lambda$, quantifies the absolute increase in growth rate caused by a marginal increase in $\rho$, given by $\mu$ itself times the proportional change, $A_\rho$. Thus, the extra term in the shadow price of transporters (compare $\theta_s$ in Eq 56 to $\theta_r$ and $\theta_e$) quantifies the growth rate benefit gained by allowing the violation of the density constraint (Eq 18) caused by a small increase in $f_s$.

Just as the economy of growth is deeply connected to protein allocation, so is growth control. For $A_j$ and $A_x$, this connection is clear from Eqs 44 and 45, respectively. For $A_\rho$, we first note that it relates to optimal marginal values via Eq 53,

$$\mu A_\rho = -\lambda = \sum_j (\partial_j \mu) f^j = \sum_s (\partial_s \mu) f^s \quad . \tag{47}$$

At optimality, the summands on the RHS are zero for the ribosome and for enzymatic reactions ($\partial_j \mu f_j = 0$ for $j = $ r, e), and the summation over $j$ can thus be restricted to only transporters $s$. Thus, at optimality, the absolute change in optimal growth rate caused by increasing $\rho$,

$\mu A_\rho$, is equal to the summed marginal effects of transport fluxes on the growth rate, $\partial_s \mu$, weighted by the flux fractions $f^s$ themselves. To see the full connection between $A_\rho$ and protein allocation, we insert Eq 33 into Eq 47 to obtain

$$A_\rho = -\sum_{l,s} \left( \frac{\partial \phi^l}{\partial f^s} \right)_\mu f^s \quad . \tag{48}$$

This equation shows that the proportional effect on the optimal growth rate that is exerted by a marginal increase in $\rho$, $A_\rho$, equals the combined marginal effects of transport fluxes $f^s$ on proteome allocation fractions, weighted by the transport fluxes themselves.

## Discussion

Modeling frameworks that are essentially linear, such as FBA and RBA, are typically analyzed numerically, as the efficiency of linear programming facilitates fast solutions even for genome-scale models [7, 8, 11]. In contrast, the construction and solution of genome-scale non-linear models faces two major obstacles, both intimately linked to the kinetic rate laws. First, experimental estimates for the required kinetic parameters—$k_{cat}$ and $K_m$ values in the simplest case of generalized Michaelis-Menten kinetics—are lacking for most reactions [42]. This problem can be alleviated by using parameter estimates from artificial intelligence approaches [43–45]. Second, the non-linearity of enzymatic rate laws makes numerical optimizations much more difficult than for linear systems, explaining why existing studies have been limited to models with only a handful of reactions [6, 15–19]. Numerical optimization is particularly problematic for models with redundant pathways, where the optimization problem is non-convex [20].

The succinct mathematical formulation for modeling balanced cellular growth developed in this paper helps to address both problems. On the one hand, the reduction of the problem description to a minimal number of independent variables—the flux fractions—reduces the dimensionality of the search space, and may thus help to accelerate numerical approaches to find optimal states. On the other hand, this formulation allowed us to identify necessary conditions for states of maximal growth rate. For enzymatic reactions $e$, these conditions (Eq 34) are "local" in the sense that they only depend on the flux fractions $\mathbf{f}$ and on the kinetic parameters of reactions directly connected to $e$ itself, i.e., they only depend on the fluxes and parameters of reactions whose turnover times $\tau^j$ are directly affected by changes in $f^e$. On the other hand, the optimality conditions for the ribosome and transport reactions (see Eqs 57 and 59) do require the knowledge of the full vector $\mathbf{f}$ and of all model parameters, which are required explicitly or via $\mu$ as determined by Eq 16.

The concise formulation also helped in the interpretation of the optimality conditions from the perspectives of economy and control theory. The marginal change in growth rate induced by each flux change is seen as the flux's marginal economic value, while the growth adaptation coefficient of each model parameter or external concentration is the change in the optimal growth rate induced by a marginal change in this parameter. The close correspondence between the mathematical expressions obtained in both perspectives helps to clarify the mathematical and conceptual links between these usually separate fields of study, including the extension of previous results of metabolic control analysis (MCA), developed for ad-hoc objectives in static sub-networks, to the holistic problem of cellular growth in GBA models. In MCA, one typically treats enzyme concentrations as control variables and studies how small changes to them affect reactant concentrations and fluxes. Here, all these variables are not only connected, but are uniquely determined by the flux fraction vector $\mathbf{f}$. Moreover, the growth rate $\mu$ itself is explicitly connected to $\mathbf{f}$ through the growth function (Eq 16). Through these connections, we can quantify the sensitivity of the cellular growth

rate, and hence approximately of organismal fitness, to changes in the control variables $\boldsymbol{\pi}$, something not possible in the usual MCA framework [9, 10]. The growth adaptation coefficients provide explicit expressions for the effects on growth rate caused by small changes in control variables at optimality. Due to the close relationship between growth rate and fitness, these estimates could be used to interpret and predict evolutionary changes in these variables.

A closely related nonlinear cellular modeling approach accounts for the different amino acid compositions of individual proteins by including "personalized" ribosome reactions for each protein [23, 46, 47]. In contrast to GBA, this type of model cannot be simplified using flux fractions **f**, as it requires a mathematical formulation that includes explicit variables for metabolite concentrations. Experimental data for *E. coli* [48] indicates that the 20 amino acid content into its total proteome changes very little over 22 highly distinct growth environments (mean coefficient of variation = 2.46%, maximal CV = 7.55%, see Table A in S1 Text), suggesting that—at least globally—different protein compositions are likely not a major factor driving significant changes in the optimal cellular state. Thus, a unique ribosome reaction with fixed column $\mathbf{M}_r$ is a realistic assumption over all these growth conditions. Further study is necessary to identify whether the different compositions of individual proteins may cause significant changes in their allocation across environments.

All analytical results in this study were derived exclusively from the growth constraints assumed in GBA models: mass conservation in balanced growth, reaction kinetics, cellular density, and non-negative concentrations. For the analysis of optimal growth, we encoded all corresponding information into a single Lagrangian function, parameterized in terms of the constraints. We formulated the problem with the flux fractions **f** as the only free variables, and used KKT conditions to obtain the necessary conditions for optimal growth states (OGSs). Through these conditions, the marginal protein allocation emerges as the natural underlying currency in the cell economy; this relationship has frequently been asserted [31, 36, 37], but is derived here entirely from first principles.

The KKT framework provides a straight-forward way to incorporate new constraints, analogous to how physical theories using the Lagrangian formalism account for additional forces by adding corresponding functions and Lagrange multipliers into the Lagrangian. A re-derivation of the KKT conditions will then result in an extended set of balance equations. Among the potential extra physiological constraints, one might consider also phenomenological constraints such as the recently reported relationship between the cellular surface/volume ratio and the growth rate [49].

One fundamental physiological limitation that could be included in this way but is not considered explicitly here is the diffusion limit of molecules within cellular compartments. This limit links density and kinetic constraints. A higher dry mass density increases the "crowding effect" within cells [26], which entails a lower diffusion rate and by consequence a longer time for reactants to find their catalysts; this effect can be modeled directly by including a corresponding dependence in the Michaelis constants $K_m$. A study on the crowding effects of all cellular concentrations—including those of small molecules—found that the observed *E. coli* dry mass density is in the range expected if evolution had optimized the cellular density for maximal growth rate [33]. In this sense, a fixed density constraint on all molecules, as considered here, may be seen as a simplifying approximation, justified by the observed constancy of cellular buoyant and dry mass densities across different growth conditions [25, 49], with the exception only of large changes in environmental osmolarities [26].

The Lagrangian formalism described here also allows a direct generalization of the theory to other objective functions, i.e., other measures of fitness at balanced growth. This can be done by incorporating a new objective function $F(\mathbf{f})$ and adding a new constraint for the

growth rate via $\omega(\mu - \mu_0)$, where $\mu$ is determined by the growth function, $\omega$ is the corresponding KKT multiplier, and $\mu_0$ is the constrained growth rate given now as an input.

An important step toward a more general theory of cellular growth would be to extend the present analytical approach to changing environments, and to derive similar analytical conditions for time-dependent optimal cellular states $\mathbf{f}(t)$. In this situation, fitness is determined by the proportional growth in a given period of time, so the objective function becomes the integral of the specific growth rate $\mu(t)$ [17, 50], under the same constraints as discussed here. This dynamical extension to a theory of proportional growth optimization would help to generalize the existing results on dynamic metabolic flux optimization [51], and building more realistic models for cells in cyclical environments, such as feast-famine cycles of the gut microbiome [52] or day-night cycles of photosynthetic microbes [19].

In sum, the concise mathematical formulation of the growth optimization problem developed here provides a powerful toolbox for the analysis and solution of mechanistic descriptions of optimal cellular physiology and growth. It thereby opens a path toward a fundamental understanding of organizing principles of biological cells. While biological systems will never be fully optimal, the study of optimal growth strategies provides an extremely useful null model for the action of natural selection.

## Methods

The necessary KKT conditions include the primal feasibility conditions given by Eqs 18 and 19), and

$$\partial_j \mathcal{L} \quad = 0 \qquad \text{(stationarity)} \tag{49}$$

$$\theta_j f_j \tau_j \quad = 0 \qquad \text{(complementary slackness)} \quad, \tag{50}$$

where $\partial_j := \partial/\partial f^j$ indicates the partial derivative with respect to $f^j$.

The stationarity conditions can be solved for the corresponding optimal multipliers $\theta_j$, resulting in

$$\theta_j = -(\partial_j \mu + \lambda \gamma_j)/\tau_j \quad, \tag{51}$$

where $\lambda$ is the optimal value for the density multiplier. After an element-wise multiplication of both sides of Eq 51 with $f_j \tau_j$, we can use the complementary slackness ($\theta_j \tau_j f_j = 0$) to get

$$(\partial_j \mu) f_j + \lambda \gamma_j f_j = 0 \quad. \tag{52}$$

Now summing the last equation over all $j$ and using the primal feasibility (Eq 18) results in

$$\lambda = -\sum_j (\partial_j \mu) f^j \quad. \tag{53}$$

Combining Eqs 51, 27 and 28, we can now express each multiplier $\theta_j$ explicitly in terms of the flux fractions $\mathbf{f}$ at optimality, resulting in slightly different expressions for ribosomal,

enzymatic, and transport reactions:

$$\theta_{\mathrm{r}} = \frac{\mu}{b^{\mathrm{a}}}\frac{1}{\tau_{\mathrm{r}}}\left(-M_{\mathrm{r}}^{\mathrm{a}} + \mu\,\tau_{\mathrm{r}} + \mu\,\mathbf{f}^{\top}\mathbf{E}_{\mathrm{r}}\right) \tag{54}$$

$$\theta_{e} = \frac{\mu}{b^{\mathrm{a}}}\frac{1}{\tau_{e}}\left(\mu\,\tau_{e} + \mu\,\mathbf{f}^{\top}\mathbf{E}_{e}\right) \tag{55}$$

$$\theta_{s} = \frac{\mu}{b^{\mathrm{a}}}\frac{1}{\tau_{s}}\left(\mu\,\tau_{s} + \mu\,\mathbf{f}^{\top}\mathbf{E}_{s} - \mu\,\mathbf{f}^{\top}\mathbf{E}\,\mathbf{f}\,\gamma_{s}\right) \quad. \tag{56}$$

By inserting these expressions into the complementary slackness conditions (Eq 50), we can now solve for $\mathbf{f}$, which results in the balance equations for ribosomal, enzymatic, and transport reactions:

$$(M_{\mathrm{r}}^{\mathrm{a}} - \mu\,\tau_{\mathrm{r}} - \mu\,\mathbf{f}^{\top}\mathbf{E}_{\mathrm{r}})f_{\mathrm{r}} = 0 \tag{57}$$

$$(\tau_{e} + \mathbf{f}^{\top}\mathbf{E}_{e})f_{e} = 0 \tag{58}$$

$$(\tau_{s} + \mathbf{f}^{\top}\mathbf{E}_{s} - \mathbf{f}^{\top}\mathbf{E}\,\mathbf{f}\,\gamma_{s})f_{s} = 0 \quad, \tag{59}$$

where we simplified the expressions by exploiting that $\mu$, $b^{\mathrm{a}}$, $\tau_{j} \neq 0$.

## Supporting information

**S1 Text. Detailed derivation of the balance equations, rate laws and kinetic parameters, mass balance and the stoichiometric matrix S, examples of GBA models, the dependence of $\lambda$ on transporters, optimal enzyme concentrations and control coefficients, Fig A (Schematics and parameters defining each model example), Table A (Amino acid frequency in the *E. coli* proteome at various growth conditions).**
(PDF)

**S1 File. Zip file containing files for numerical optimization.** As described in the S1 Text.
(ZIP)

## Acknowledgments

We thank Stefan Müller for discussions about KKT conditions, Alexander Kroll for verifying early calculations, and Xiao-Pan Hu for providing data in Table A in S1 Text.

## Author Contributions

**Conceptualization:** Hugo Dourado.

**Formal analysis:** Hugo Dourado, Wolfram Liebermeister.

**Funding acquisition:** Oliver Ebenhöh, Martin J. Lercher.

**Investigation:** Hugo Dourado.

**Methodology:** Hugo Dourado.

**Project administration:** Martin J. Lercher.

**Software:** Hugo Dourado.

**Supervision:** Oliver Ebenhöh, Martin J. Lercher.

**Visualization:** Hugo Dourado.

**Writing – original draft:** Hugo Dourado, Wolfram Liebermeister, Martin J. Lercher.

**Writing – review & editing:** Hugo Dourado, Wolfram Liebermeister, Oliver Ebenhöh, Martin J. Lercher.

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
