## [Decision Letter · Decision Letter 0]

30 Jan 2023

Dear Dr. Dourado,

Thank you very much for submitting your manuscript "Growth Mechanics: General principles of optimal cellular resource allocation in balanced growth" for consideration at PLOS Computational Biology.

As with all papers reviewed by the journal, your manuscript was reviewed by members of the editorial board and by several independent reviewers. In light of the reviews (below this email), we would like to invite the resubmission of a significantly-revised version that takes into account the reviewers' comments. Please note that the correct Reviewer #3 comments are now attached.

While you should address all reviewer's comments, the most important issue would be to provide a more extensive biological motivation for the work and a case study, as requested by Reviewer 1.

We cannot make any decision about publication until we have seen the revised manuscript and your response to the reviewers' comments. Your revised manuscript is also likely to be sent to reviewers for further evaluation.

Sincerely,

Kiran Raosaheb Patil, Ph.D.

Section Editor

PLOS Computational Biology

Kiran Patil

Section Editor

PLOS Computational Biology

Reviewer's Responses to Questions

**Comments to the Authors:**

Reviewer #1: Summary:

In this work, the authors introduce a mathematical method to perform growth balance analysis based on a known method to solve convex nonlinear optimization. The current manuscript only focuses on the mathematical formulation but does not provide biologically relevant examples to demonstrate the usefulness of the formulation. The manuscript often reads as lacking in essential detail. Comments are also provided below after the reviewing process. In addition, the reviewer suggests that the authors move detailed mathematical derivations and formulation to Supplementary Methods and focus on explaining the formulation (i.e., so as to help readers understand rather than derive the formulation).

Major concerns

The introduction appears to present only a partial picture of the proposed GM framework with respect to other, non-linear, metabolic modeling frameworks, where many are concerned with the paucity of information needed to construct a kinetic model. Frameworks such as Ensemble Models (EMs), GRASP, iSCHRUNK (within ORACLE), and K-FIT are used to overcome this barrier. Is this not also a barrier to the GBA/GM frameworks? Other common terms in linear modeling frameworks, such as genome-scale models (used in the discussion, line 590) should be introduced here if used elsewhere.

Do elements of vector v in M*v = mu*c (Eq. 3) correspond to fluxes of the reactions facilitated/catalyzed by transporters (transport reaction), enzymes (metabolic reaction), and ribosomes (protein translation reaction)? Or do those elements in v correspond to those machineries’ synthesis fluxes? If the former is true, then how are the machiney’s synthesis fluxes accounted for in the formulation?

In addition, it is unclear to the reviewer on what constraints are formed from the column “r” and row “a” of M. On protein and ribosome, total protein concentration is constrained but to be less than a variable “c”? Then, “c^a” is constrained to be less than “rho”. Is “rho” a constant/parameter? What about ribosome concentration? Does this mean the model’s maximal growth rate is limited by other constraint(s) rather than protein and ribosome availability (as proteins and ribosomes can be produced as much as the cells need)?

Line 183-184, the statement “The property (2) guarantees mass conservation within reactions…” is not true as M is derived from S. The two sets of positive and negative coefficients in M were mentioned to be normalized on individual reactions. In other words, the two sums of molecular weight of positive and negative terms, respectively, can differ (which indicate reaction is mass, and elemental, imbalances).

To create Eq. 11, does this mean that the growth rate “mu” has to be constant? Thus, to solve a GBA optimization (previous or current formulation), does “mu” need to be set to a constant? If so, how could one solve for the maximal “mu” in GBA?

Line 222, the authors state that extracellular concentrations, “x”, are constant. Is this assumption compatible with simulation for batch conditions (i.e., nutrients are depleted over time)?

Based on Eq. 19, are flux fraction “f_j” variables positive? What about reversible reactions? Also, could the Eq. 19 constraint be replaced by the “f_j >= 0” constraint because “tau_j” are positive? If not, could the authors provide a comment on the current form of Eq. 19?

It would be helpful to readers to include in the growth problem (Eq. 21) notation to indicate the variables’ domains (e.g., are flux variables strictly non-negative?). This could also be helpful in understanding how to construct models which can use this framework (e.g. are reversible reactions allowed?).

The reviewer does not find support for the statement presented in lines 605 to 607: “These conditions are local for each reaction, i.e., they do not require complete knowledge of the cellular reaction network and its kinetics.” Without complete knowledge, how are M and “tau” defined? In this work, overall, there is no concrete, concise, statement or statements as to what knowledge is needed to build a model to which this analysis can be applied.

According to the abstract and title, the purpose of this manuscript is to introduce the Growth Mechanics (GM) framework, but the phrase “Growth Mechanics” is only used in the title and abstract, and the acronym is only used in the abstract and last paragraph of the discussion. What is the GM framework? How is it different from the GBA framework, and where are the generalizations which are mentioned in the abstract? How is it more powerful? Some discussion on the distinctiveness of the two frameworks and a comparison of their applications would be useful.

Marked inconsistencies exist in how “tau” is discussed. In Table 1, it is noted as a turnover time for the reaction, and later on discussed as the inverse of the usual factor in kinetic rate laws (lines 152 to 153, the reviewer assumed this refers to a “k_cat” with units of inverse time, so it is understood that “k_cat” = 1/“tau”). In enzyme kinetics, turnover number is a constant for the system (e.g. “k_cat” = “V_max” / “E_t”). So Table 1 and the beginning of the manuscript suggest a constant “tau” value. However, In line 148, this statement is made: “...and adding the kinetic rate laws ‘tau’ and density ‘rho’”. Further, in lines 156 to 158, it is stated that “tau” is dependent on concentrations for both participant and non-participant metabolites. In Eq. 4, τ is treated as a concentration-dependent function. In these lines and equations, it appears that “tau” evolves from a simple turnover time vector into a Michaelis-Menten form description of the reaction rate, as lines 161 to 165 discuss Michaelis constants and turnover numbers, as well as the unit for input to this equation, and line 333 discusses using a “tau” that follows “a simple Michaelis-Menten rate law”.

How are proteins included and formulated in the model? Is it then that there is a single pseudo-protein in the GBA models (the section from lines 118 to 120 is written in the singular form, indicating a single protein metabolite and single protein-producing reaction from the ribosome)? How is the composition of this single protein determined? Note that this seems to contradict lines 149 to 151, Eqs. 4 and 5, as well as many other places in the manuscript which suggest that each reaction j has an associated protein or protein complex.

This manuscript is considered by the reviewer to be incomplete because no (biologically) relevant examples and applications are provided for the methods and computational models. The authors mentioned examples in Supplementary Materials but they are only simple toy examples.

If the tool requires the use of pre-defined kinetic law (in “tau”) with pre-defined kinetic parameters, then some other tool would need to be used in conjunction with this modeling framework. This is acknowledged in Lines 592 to 595. However, many such parameter estimating tools or approaches (such as ensemble modeling) integrate parameter estimation with kinetic parameter estimation. Therefore, what would be the advantage of using this approach as opposed to an integrated approach? How do these model structures compare with other kinetic modeling approaches?

It appears that a number of bilinear terms exist in the formulation. How is this consistent with the purported convexity of the formulation?

Minor concerns

Line 15, Constraint Based Reconstruction and Analysis (COBRA) and Genome-Scale Model (GSM or GEM) are more commonly used acronyms, and should be included here for clarity and easier linking to related works.

Lines 33-40, is the argument being made that RBA and ME models are both CBM models? If so, specify. Many other works consider these a different class of models from GSM models, so this could be unintentionally confusing if not clarified.

Lines 43-44, “widely adopted” rather than “most powerful” should be used.

Table 1, please make sure that a symbol is defined before it is used (index i is noted as containing m and a, but m is not yet defined).

Line 110, for clarity please specify if M,τ, and ρ are parameters or variables.

Lines 123-124, please specify if the negative and positive entries are within a particular column.

Line 163, please ensure consistency in how units are described. In Table 1, the units for v are described as “[mass][volume]^-1[time]^-1” whereas here they are described as “[mass x volume ^-1 x time ^-1]”. This happens elsewhere as well, this example is intended to draw all such instances to the authors’ attention.

Line 168, could the value for “rho” be provided in the main text?

Line 237, please specify the “the previous two equations”.

Reviewer #2: Review of Growth Mechanics: General principles of optimal cellular resource allocation in balanced growth

by Bob Planqué

This is a well-written and most welcome paper on the conditions that hold in states of optimal balanced growth, in the particular case of an EFM/EGM, and in which it is assumed that all protein and ribosome in the cell may be lumped into one protein compartment. I particularly like the new view of reformulating the balanced growth equations using flux fractions, and the fact that in this particular model the metabolite concentration indeed drop out of the equations, even though they were explicitly taken into account to start with.

I have enjoyed reading the manuscript, and only have a number of comments for improvement, clarification and link to other literature.

One of the omissions that I think should really be dealt with is the link to the elementary mode literature. The case considered is that in which the reaction matrix has full rank. This is equivalent to restricting to an Elementary Mode, whether it is an Elem. Flux Mode [refs 21 ,22 in the ms] or an Elem Growth Mode [ref 44,45, maybe also 46?], as I am sure the authors are well aware. Given the central role E(F/G)Ms have come to play in our systems biology literature, I think it is essential that this concept is mentioned, rather than just refer to papers that deal with them.

In l 87, the authors refer to two papers in which it is proved that EFMs are specific flux optimisers. In [44], it is shown that EGMs are growth rate optimisers. The model considered in this paper falls somewhat in the middle of these approaches (EFMs disregarding the self-replication aspect and not accounting for protein synthesis, EGMs accounting for everything, differentiating between different enzyme synthesis rates, etc.), and it is not immediately clear to me that the EFM proofs apply to the case at hand. Maybe it is best to refer to [44] (and maybe also [45] which deals with other cases in-between EFMs and EGMs and also contains proofs of growth rate optimisation in elementary modes) when making the claim that one may restrict to the case of full matrix rank. Then all bases are covered.

l 117 and further: Here the matrix M is introduced. I had trouble understanding the construction of the last row of M, even though it turned out to be trivial. I do not think the construction follows Molenaar et al (who did differentiate between protein compartments, and were the inspiration of the introduction of the 'alpha' ribosomal fractions used in [44]). Please explain this last row more clearly, preferably by giving a tiny example. At present, examples are in the SI (but this is not mentioned in the ms at this point, so this could be another solution), which is not too accessible to the reader.

In Eq (5) - (6) there are two constraints, but in l 192 it is mentioned that the protein constraint is an emergent property, rather than a real hard constraint (i.e. c^a is not known beforehand). So why include it then? I guess I'm missing something here.

l 242: the recursion has been noted in older papers, starting with the RBA models by Goelzer et al (2009). It is a mainstay of ME-type models. It is clearly also mentioned in [44]. The insight that this recursion disappears here is very neat, and I need to think about that more deeply. But please add some refs here.

Growth analysis, page 9.

This whole section aims to derive conditions that hold at optimality + steady state. In particular for EFMs (without the self-replication part taken into account), this has been done in Planqué et al. (2018), in which such equations are coupled to dynamic enzyme synthesis rates to control the maximal specific flux in varying environments. The situation here is of course a bit different, but the two situations are closely related, so a reference seems in place, either here or in the Discussion.

In this section, it is also not mentioned whether such optimal states actually necessarily exist, or whether there are multiple (local) optima. This all has to do with the convexity properties of the relevant functional, of which the authors are well aware. References such as the paper by Wolfram and Elad in 2016 on Enzyme Cost Minimisation and convexity, and also (Planqué et al. 2018) which improves this slightly to strict convexity, are relevant, but I don't think they solve the case here immediately. The authors would do well to change 'the optimal state' to 'an optimal state' in several places, such as in line 345 and 385.

Discussion

l 606: I read there are local conditions for each reaction as necessary conditions for optimal growth. I didn't quite understand this part of the ms, I have to say (I didn't have the time to think in detail about it), but I find this surprising. Surely, because reactions have substrates and products, such conditions must be coupled? See Planqué et al (2018) for a situation where this is clearly the case. But as I said, the situations do not exactly compare.

I somehow find the idea of lumping all protein and ribosome into one pool, while on the other hand calculating (differing) steady state protein and ribosome concentrations for different conditions peculiar. Would it be not more natural to explain this by saying that all protein synthesis rates (per unit of ribosome) are assumed to be equal (which is also what comes out of having a constant relevant amino acid abundance assumption, see l 638 and which is also discussed in [44])? Now it sometimes reads as if you both lump things, and not lump things. The idea of marginal protein allocation clearly hinges on proteins being present in different concentrations (in optimum or otherwise). I think this just needs to be clarified, preferably already in the part where the model is introduced.

l 683: The situation considered here is that of optimal control, but there are alternatives, such as adaptive control. In the latter case, this is essentially the qORAC like framework, see Planqué et al (2018). As it already exists (for a slightly different case than the one considered here), and the extension from qORAC to the present paper would be only a small change (I think), it should be cited here.

Minor comments and questions:

l 36: the burden => the enzyme/protein burden?

l 72: what is the difference between a fixed protein concentration and a fixed combined mass density of their components? I think I understand because I know how this is usually formulated, but the reader might not.

l 284: set OF kinetic parameters

l 294: what are shadow prices? I understand there must be some link to economic arguments, but this needs to be explained, or at least given some reference to aid the reader.

References:

Planqué et al. (2018).  Maintaining maximal metabolic rates by gene expression control. PLoS Comp Biol. 14(9):e1006412.

Reviewer #3: In this work, Dourado et al. extend their framework called Growth Balance Analysis to a more general system with several meaningful advantages. All formulae are expressed using a single independent variable f (for flux) and therefore are simpler than other types of cell growth models. By this, they lay the groundwork for a universal modeling approach which has the potential to bring together many disjoint approaches and hopefully increase cooperation between modelers that use FBA, MCA, kinetic models and others. The text is very well written and I haven't found any scientific or mathematical issues.

I can offer a few suggestions that might improve the text even further:

1. The use of Einstein's summation convention is, in my view, not very helpful. Indeed, it might appeal to some physicists and could be a bit less verbose - but I don't think the benefit outweighs the downside of being less standard and harder to read for many people.

2. I might have misunderstood, but it appears that fluxes (v^j) and turnover times (τ^j) can be negative (and indeed they are not constrained to be non-negative like other variables). I think stating this explicitly could help readability.

3. At some point, metabolite concentrations and protein concentrations are "replaced" with fluxes based on the balanced growth assumption (i.e. unlike in FBA where fluxes balance to 0, here each one balances to the dilution rate of the metabolite/protein defined by cellular growth). First, this idea was presented in a similar context in 2010 by Benyamini et al. (https://doi.org/10.1186/gb-2010-11-4-r43), albeit only for metabolites.

Furthermore, in the discussion (lines 602-603) the authors highlight the advantage of using this approach in minimizing the number of independent variables and thus assisting the numerical solvers. However, this might be a slight overstatement since many models indeed have explicit protein concentration variables, but then the biosynthesis flux is the dependent variable (and very simple to express as a function of the concentration). For metabolites, the case is not very different (the Sv = 0 constraint might have more rows and be more rank deficient, but almost all solvers can deal with this easily).

4. The manuscript is quite long and dense (in terms of mathematical definitions and derivations). There is no easy solution to this, but perhaps the authors might consider splitting it or moving some parts to a supplementary section and really focus only on the main message. In addition, perhaps a few toy examples (with simulations or analytical solutions) could be helpful as keeping track of all the abstract math symbols all the way until the end is a bit daunting. That being said, I greatly appreciated the table of symbols on page 4. Perhaps one could add more of these symbols to figure 1A as well?

**Have the authors made all data and (if applicable) computational code underlying the findings in their manuscript fully available?**

Reviewer #1: Yes

Reviewer #2: Yes

Reviewer #3: Yes

PLOS authors have the option to publish the peer review history of their article (what does this mean?). If published, this will include your full peer review and any attached files.

Reviewer #1: No

Reviewer #2: No

Reviewer #3: **Yes: **Elad Noor
---

## [Decision Letter · Decision Letter 1]

4 May 2023

Dear Dr. Dourado,

We are pleased to inform you that your manuscript 'Mathematical properties of optimal fluxes in cellular reaction networks at balanced growth' has been provisionally accepted for publication in PLOS Computational Biology.

Best regards,

Pedro Mendes, PhD

Academic Editor

PLOS Computational Biology

Kiran Patil

Section Editor

PLOS Computational Biology

Reviewer's Responses to Questions

**Comments to the Authors:**

Reviewer #1: The authors have addressed our comments and suggestions for changes. The revised manuscript is significantly improved.

Reviewer #3: The authors have addressed all of my and the other reviewers' comments.

**Have the authors made all data and (if applicable) computational code underlying the findings in their manuscript fully available?**

Reviewer #1: Yes

Reviewer #3: Yes

PLOS authors have the option to publish the peer review history of their article (what does this mean?). If published, this will include your full peer review and any attached files.

Reviewer #1: No

Reviewer #3: **Yes: **Elad Noor

---

## [Editor Report · Acceptance letter]

1 Jun 2023

PCOMPBIOL-D-22-01649R1 

Mathematical properties of optimal fluxes in cellular reaction networks at balanced growth

Dear Dr Dourado,

I am pleased to inform you that your manuscript has been formally accepted for publication in PLOS Computational Biology. Your manuscript is now with our production department and you will be notified of the publication date in due course.

With kind regards,

Zsofia Freund
